# Offline Neural Contextual Bandits: Pessimism, Optimization and Generalization

**Thanh Nguyen-Tang** *
Applied AI Institute
Deakin University

**Sunil Gupta**
Applied AI Institute
Deakin University

**A. Tuan Nguyen**
Department of Engineering Science
University of Oxford

**Svetha Venkatesh**
Applied AI Institute
Deakin University

## Abstract

Offline policy learning (OPL) leverages existing data collected a priori for policy optimization without any active exploration. Despite the prevalence and recent interest in this problem, its theoretical and algorithmic foundations in function approximation settings remain under-developed. In this paper, we consider this problem on the axes of distributional shift, optimization, and generalization in offline contextual bandits with neural networks. In particular, we propose a provably efficient offline contextual bandit with neural network function approximation that does not require any functional assumption on the reward. We show that our method provably generalizes over unseen contexts under a milder condition for distributional shift than the existing OPL works. Notably, unlike any other OPL method, our method learns from the offline data in an online manner using stochastic gradient descent, allowing us to leverage the benefits of online learning into an offline setting. Moreover, we show that our method is more computationally efficient and has a better dependence on the effective dimension of the neural network than an online counterpart. Finally, we demonstrate the empirical effectiveness of our method in a range of synthetic and real-world OPL problems.

## 1 Introduction

We consider the problem of offline policy learning (OPL) (Lange et al., 2012; Levine et al., 2020) where a learner infers an optimal policy given only access to a fixed dataset collected a priori by unknown behaviour policies, without any active exploration. There has been growing interest in this problem recently, as it reflects a practical paradigm where logged experiences are abundant but an interaction with the environment is often limited, with important applications in practical settings such as healthcare (Gottesman et al., 2019; Nie et al., 2021), recommendation systems (Strehl et al., 2010; Thomas et al., 2017), and econometrics (Kitagawa & Tetenov, 2018; Athey & Wager, 2021).

Despite the importance of OPL, theoretical and algorithmic progress on this problem has been rather limited. Specifically, most existing works are restricted to a strong parametric assumption of environments such as tabular representation (Yin & Wang, 2020; Buckman et al., 2020; Yin et al., 2021; Yin & Wang, 2021; Rashidinejad et al., 2021; Xiao et al., 2021) and more generally as linear models (Duan & Wang, 2020; Jin et al., 2020; Tran-The et al., 2021). However, while the linearity assumption does not hold for many problems in practice, no work has provided a theoretical guarantee and a practical algorithm for OPL with neural network function approximation.

In OPL with neural network function approximation, three fundamental challenges arise:

**Distributional shift**. As OPL is provided with only a fixed dataset without any active exploration, there is often a mismatch between the distribution of the data generated by a target policy and that of the offline data. This distributional mismatch can cause erroneous value overestimation and render

---
*Email: nguyent2792@gmail.com.

many standard online policy learning methods unsuccessful (Fujimoto et al., 2019). To guarantee an efficient learning under distributional shift, common analyses rely on a sort of uniform data coverage assumptions (Munos & Szepesvári, 2008; Chen & Jiang, 2019; Brandfonbrener et al., 2021; Nguyen-Tang et al., 2021) that require the offline policy to be already sufficiently explorative over the entire state space and action space. To mitigate this strong assumption, a pessimism principle that constructs a lower confidence bound of the reward functions for decision-making (Rashidinejad et al., 2021) can reduce the requirement to a single-policy concentration condition that requires the coverage of the offline policy only on the target policy. However, Rashidinejad et al. (2021) only uses this condition for tabular representation and it is unclear whether complex environments such as ones that require neural network function approximation can benefit from this condition. Moreover, the single-policy concentration condition still requires the offline policy to be stationary (e.g., the actions in the offline data are independent and depend only on current state). However, this might not hold for many practical scenarios, e.g., when the offline data was collected by an active learner (e.g., by an Q-learning algorithm). Thus, it remains unclear what is a minimal structural assumption on the distributional shift that allows a provably efficient OPL algorithm.

**Optimization**. Solving OPL often involves in fitting a model into the offline data via optimization. Unlike in simple function approximations such as tabular representation and linear models where closed-form solutions are available, OPL with neural network function approximation poses an additional challenge that involves a non-convex, non-analytical optimization problem. However, existing works of OPL with function approximation ignore such optimization problems by assuming free access to an optimization oracle that can return a global minimizer (Brandfonbrener et al., 2021; Duan et al., 2021; Nguyen-Tang et al., 2021) or an empirical risk minimizer with a pointwise convergence bound at an exponential rate (Hu et al., 2021). This is largely not the case in practice, especially in OPL with neural network function approximation where a model is trained by gradient-based methods such as stochastic gradient descents (SGD). Thus, to understand OPL in more practical settings, it is crucial to consider optimization in design and analysis. To our knowledge, such optimization problem has not been studied in the context of OPL with neural network function approximation.

**Generalization**. In OPL, generalization is the ability to generalize beyond the states (or contexts as in the specific case of stochastic contextual bandits) observed in the offline data. In other words, an offline policy learner with good generalization should obtain high rewards not only in the observed states but also in the entire (unknown) state distribution. The challenge of generalization in OPL is that as we learn from the fixed offline data, the learned policy has highly correlated structures where we cannot directly use the standard concentration inequalities (e.g. Hoeffding's inequality, Bernstein inequality) to derive a generalization bound. The typical approaches to overcome this difficulty are data splitting and uniform convergence. While data splitting splits the offline data into disjoint folds to break the correlated structures (Yin & Wang, 2020), uniform convergence establishes generalization uniformly over a class of policies learnable by a certain model (Yin et al., 2021). However, in the setting where the offline data itself can have correlated structures (e.g., an offline action can depend on the previous offline data) and the model used is sufficiently large that renders a uniform convergence bound vacuous, neither the data splitting technique in (Yin & Wang, 2020) nor the uniform convergence argument in (Yin et al., 2021) yield a good generalization. Thus, it is highly non-trivial to guarantee a strong generalization in OPL with neural network function approximation from highly correlated offline data.

In this paper, we consider the problem of OPL with neural network function approximation on the axes of distributional shift, optimization and generalization via studying the setting of stochastic contextual bandits with overparameterized neural networks. Specifically, we make three contributions toward enabling OPL in more practical settings:

- First, we propose an algorithm that uses a neural network to model any bounded reward function without assuming any functional form (e.g., linear models) and uses a pessimistic formulation to deal with distributional shifts. Notably, unlike any standard offline learning methods, our algorithm learns from the offline data in an online-like manner, allowing us to leverage the advantages of online learning into offline setting.

- Second, our theoretical contribution lies in making the generalization bound of OPL more realistic by taking into account the optimization aspects and requiring only a milder condition for distributional shifts. In particular, our algorithm uses stochastic gradient descent and updates the network completely online, instead of retraining it from scratch for every

iteration. Moreover, the distributional shift condition in our analysis, unlike in the existing works, does not require the offline policy to be uniformly explorative or stationary. Specifically, we prove that, under mild conditions with practical considerations above, our algorithm learns the optimal policy with an expected error of $\tilde{\mathcal{O}}(\kappa \tilde{d}^{1/2} n^{-1/2})$, where $n$ is the number of offline samples, $\kappa$ measures the distributional shift, and $\tilde{d}$ is an effective dimension of the neural network that is much smaller than the network capacity (e.g., the network's VC dimension and Rademacher complexity).

- Third, we evaluate our algorithm in a number of synthetic and real-world OPL benchmark problems, verifying its empirical effectiveness against the representative methods of OPL.

**Notation**. We use lower case, bold lower case, and bold upper case to represent scalars, vectors and matrices, respectively. For a vector $\boldsymbol{v} = [v_1, \ldots, v_d]^T \in \mathbb{R}^d$ and $p > 1$, denote $\|\boldsymbol{v}\|_p = (\sum_{i=1}^d v_i^p)^{1/p}$ and let $[\boldsymbol{v}]_j$ be the $j^{\text{th}}$ element of $\boldsymbol{v}$. For a matrix $\boldsymbol{A} = (A_{i,j})_{m \times n}$, denote $\|\boldsymbol{A}\|_F = \sqrt{\sum_{i,j} A_{i,j}^2}$, $\|\boldsymbol{A}\|_p = \max_{\boldsymbol{v}:\|\boldsymbol{v}\|_p=1} \|\boldsymbol{A}\boldsymbol{v}\|_p$, $\|\boldsymbol{A}\|_\infty = \max_{i,j} |A_{i,j}|$ and let $\text{vec}(\boldsymbol{A}) \in \mathbb{R}^{mn}$ be the vectorized representation of $\boldsymbol{A}$. For a square matrix $\boldsymbol{A}$, a vector $\boldsymbol{v}$, and a matrix $\boldsymbol{X}$, denote $\|\boldsymbol{v}\|_{\boldsymbol{A}} = \sqrt{\boldsymbol{v}^T \boldsymbol{A} \boldsymbol{v}}$ and $\|\boldsymbol{X}\|_{\boldsymbol{A}} = \|\text{vec}(\boldsymbol{X})\|_{\boldsymbol{A}}$. For a collection of matrices $\boldsymbol{W} = (\boldsymbol{W}_1, \ldots, \boldsymbol{W}_L)$ and a square matrix $\boldsymbol{A}$, denote $\|\boldsymbol{W}\|_F = \sqrt{\sum_{l=1}^L \|\boldsymbol{W}_l\|_F^2}$, and $\|\boldsymbol{W}\|_{\boldsymbol{A}} = \sqrt{\sum_{l=1}^L \|\boldsymbol{W}_l\|_{\boldsymbol{A}}^2}$. For a collection of matrices $\boldsymbol{W}^{(0)} = (\boldsymbol{W}_1^{(0)}, \ldots, \boldsymbol{W}_L^{(0)})$, denote $\mathcal{B}(\boldsymbol{W}^{(0)}, R) = \{\boldsymbol{W} = (\boldsymbol{W}_1, \ldots, \boldsymbol{W}_L) : \|\boldsymbol{W}_l - \boldsymbol{W}_l^{(0)}\|_F \le R\}$. Denote $[n] = \{1, 2, \ldots, n\}$, and $a \vee b = \max\{a, b\}$. We write $\tilde{\mathcal{O}}(\cdot)$ to hide logarithmic factors in the standard Big-Oh notation, and write $m \ge \Theta(f(\cdot))$ to indicate that there is an absolute constant $C > 0$ that is independent of any problem parameters $(\cdot)$ such that $m \ge Cf(\cdot)$.

## 2 BACKGROUND

In this section, we provide essential background on offline stochastic contextual bandits and overparameterized neural networks.

### 2.1 STOCHASTIC CONTEXTUAL BANDITS

We consider a stochastic $K$-armed contextual bandit where at each round $t$, an online learner observes a full context $\boldsymbol{x}_t := \{\boldsymbol{x}_{t,a} \in \mathbb{R}^d : a \in [K]\}$ sampled from a context distribution $\rho$, takes an action $a_t \in [K]$, and receives a reward $r_t \sim P(\cdot|\boldsymbol{x}_{t,a_t})$. A policy $\pi$ maps a full context (and possibly other past information) to a distribution over the action space $[K]$. For each full context $\boldsymbol{x} := \{\boldsymbol{x}_a \in \mathbb{R}^d : a \in [K]\}$, we define $v^\pi(\boldsymbol{x}) = \mathbb{E}_{a \sim \pi(\cdot|\boldsymbol{x}), r \sim P(\cdot|\boldsymbol{x}_a)}[r]$ and $v^*(\boldsymbol{x}) = \max_\pi v^\pi(\boldsymbol{x})$, which is attainable due to the finite action space.

In the offline contextual bandit setting, the goal is to learn an optimal policy only from an offline data $\mathcal{D}_n = \{(\boldsymbol{x}_t, a_t, r_t)\}_{t=1}^n$ collected a priori by a behaviour policy $\mu$. The goodness of a learned policy $\hat{\pi}$ is measured by the (expected) sub-optimality the policy achieves in the entire (unknown) context distribution $\rho$:

$$\text{SubOpt}(\hat{\pi}) := \mathbb{E}_{\boldsymbol{x} \sim \rho}[\text{SubOpt}(\hat{\pi}; \boldsymbol{x})], \text{ where } \text{SubOpt}(\hat{\pi}; \boldsymbol{x}) := v^*(\boldsymbol{x}) - v^{\hat{\pi}}(\boldsymbol{x}).$$

In this work, we make the following assumption about reward generation: For each $t$, $r_t = h(\boldsymbol{x}_{t,a_t}) + \xi_t$, where $h : \mathbb{R}^d \to [0, 1]$ is an unknown reward function, and $\xi_t$ is a $R$-subgaussian noise conditioned on $(\mathcal{D}_{t-1}, \boldsymbol{x}_t, a_t)$ where we denote $\mathcal{D}_t = \{(\boldsymbol{x}_\tau, a_\tau, r_\tau)\}_{1 \le \tau \le t}, \forall t$. The $R$-subgaussian noise assumption is standard in stochastic bandit literature (Abbasi-Yadkori et al., 2011; Zhou et al., 2020; Xiao et al., 2021) and is satisfied e.g. for any bounded noise.

### 2.2 OVERPARAMETERIZED NEURAL NETWORKS

To learn the unknown reward function without any prior knowledge about its parametric form, we approximate it by a neural network. In this section, we define the class of overparameterized neural networks that will be used throughout this paper. We consider fully connected neural networks with depth $L \ge 2$ defined on $\mathbb{R}^d$ as

$$f_{\boldsymbol{W}}(\boldsymbol{u}) = \sqrt{m} \boldsymbol{W}_L \sigma \left(\boldsymbol{W}_{L-1} \sigma \left(\ldots \sigma(\boldsymbol{W}_1 \boldsymbol{u}) \ldots\right)\right), \forall \boldsymbol{u} \in \mathbb{R}^d, \tag{1}$$

---

**Algorithm 1 NeuraLCB**

---

**Input:** Offline data $\mathcal{D}_n = \{(\boldsymbol{x}_t, a_t, r_t)\}_{t=1}^n$, step sizes $\{\eta_t\}_{t=1}^n$, regularization parameter $\lambda > 0$, confidence parameters $\{\beta_t\}_{t=1}^n$.

1: Initialize $\boldsymbol{W}^{(0)}$ as follows: set $\boldsymbol{W}_l^{(0)} = [\bar{\boldsymbol{W}}_l, \ \boldsymbol{0}; \boldsymbol{0}, \ \bar{\boldsymbol{W}}_l], \forall l \in [L-1]$ where each entry of $\bar{\boldsymbol{W}}_l$ is generated independently from $\mathcal{N}(0, 4/m)$, and set $\boldsymbol{W}_L^{(0)} = [\boldsymbol{w}^T, -\boldsymbol{w}^T]$ where each entry of $\boldsymbol{w}$ is generated independently from $\mathcal{N}(0, 2/m)$.

2: $\boldsymbol{\Lambda}_0 \leftarrow \lambda \boldsymbol{I}$.

3: **for** $t = 1, \ldots, n$ **do**

4:      Retrieve $(\boldsymbol{x}_t, a_t, r_t)$ from $\mathcal{D}_n$.

5:      $\hat{\pi}_t(\boldsymbol{x}) \leftarrow \arg\max_{a \in [K]} L_t(\boldsymbol{x}_a)$, for all $\boldsymbol{x} = \{\boldsymbol{x}_a \in \mathbb{R}^d : a \in [K]\}$ where $L_t(\boldsymbol{u}) = f_{\boldsymbol{W}^{(t-1)}}(\boldsymbol{u}) - \beta_{t-1}\|\nabla f_{\boldsymbol{W}^{(t-1)}}(\boldsymbol{u}) \cdot m^{-1/2}\|_{\boldsymbol{\Lambda}_{t-1}^{-1}}, \forall \boldsymbol{u} \in \mathbb{R}^d$

6:      $\boldsymbol{\Lambda}_t \leftarrow \boldsymbol{\Lambda}_{t-1} + \text{vec}(\nabla f_{\boldsymbol{W}^{(t-1)}}(\boldsymbol{x}_{t,a_t})) \cdot \text{vec}(\nabla f_{\boldsymbol{W}^{(t-1)}}(\boldsymbol{x}_{t,a_t}))^T/m$.

7:      $\boldsymbol{W}^{(t)} \leftarrow \boldsymbol{W}^{(t-1)} - \eta_t \nabla \mathcal{L}_t(\boldsymbol{W}^{(t-1)})$ where $\mathcal{L}_t(\boldsymbol{W}) = \frac{1}{2}(f_{\boldsymbol{W}}(\boldsymbol{x}_{t,a_t}) - r_t)^2 + \frac{m\lambda}{2}\|\boldsymbol{W} - \boldsymbol{W}^{(0)}\|_F^2$.

8: **end for**

**Output:** Randomly sample $\hat{\pi}$ uniformly from $\{\hat{\pi}_1, \ldots, \hat{\pi}_n\}$.

---

where $\sigma(\cdot) = \max\{\cdot, 0\}$ is the rectified linear unit (ReLU) activation function, $\boldsymbol{W}_1 \in \mathbb{R}^{m \times d}, \boldsymbol{W}_i \in \mathbb{R}^{m \times m}, \forall i \in [2, L-1], \boldsymbol{W}_L \in \mathbb{R}^{m \times 1}$, and $\boldsymbol{W} := (\boldsymbol{W}_1, \ldots, \boldsymbol{W}_L)$ with $\text{vec}(\boldsymbol{W}) \in \mathbb{R}^p$ where $p = md + m + m^2(L-2)$. We assume that the neural network is overparameterized in the sense that the width $m$ is sufficiently larger than the number of samples $n$. Under such an overparameterization regime, the dynamics of the training of the neural network can be captured in the framework of so-called neural tangent kernel (NTK) (Jacot et al., 2018). Overparameterization has been shown to be effective to study the interpolation phenomenon and neural training for deep neural networks (Arora et al., 2019; Allen-Zhu et al., 2019; Hanin & Nica, 2019; Cao & Gu, 2019; Belkin, 2021).

## 3 ALGORITHM

In this section, we present our algorithm, namely NeuraLCB (which stands for **Neura**l **L**ower **C**onfidence **B**ound). A key idea of NeuraLCB is to use a neural network $f_{\boldsymbol{W}}(\boldsymbol{x}_a)$ to learn the reward function $h(\boldsymbol{x}_a)$ and use a pessimism principle based on a lower confidence bound (LCB) of the reward function (Buckman et al., 2020; Jin et al., 2020) to guide decision-making. The details of NeuraLCB are presented in Algorithm 1. Notably, unlike any other OPL methods, NeuraLCB learns in an online-like manner. Specifically, at step $t$, Algorithm 1 retrieves $(\boldsymbol{x}_t, a_t, r_t)$ from the offline data $\mathcal{D}_n$, computes a lower confidence bound $L_t$ for each context and action based on the current network parameter $\boldsymbol{W}^{(t-1)}$, extracts a greedy policy $\hat{\pi}_t$ with respect to $L_t$, and updates $\boldsymbol{W}^{(t)}$ by minimizing a regularized squared loss function $\mathcal{L}_t(\boldsymbol{W})$ using stochastic gradient descent. Note that Algorithm 1 updates the network using one data point at time $t$, does not use the last sample $(\boldsymbol{x}_n, a_n, r_n)$ for decision-making and takes the average of an ensemble of policies $\{\hat{\pi}_t\}_{t=1}^n$ as its returned policy. These are merely for the convenience of theoretical analysis. In practice, we can either use the ensemble average, the best policy among the ensemble or simply the latest policy $\hat{\pi}_n$ as the returned policy. At step $t$, we can also train the network on a random batch of data from $\mathcal{D}_t$ (the "B-mode" variant as discussed in Section 6).

## 4 GENERALIZATION ANALYSIS

In this section, we analyze the generalization ability of NeuraLCB. Our analysis is built upon the neural tangent kernel (NTK) (Jacot et al., 2018). We first define the NTK matrix for the neural network function in Eq. (1).

**Definition 4.1** (Jacot et al. (2018); Cao & Gu (2019); Zhou et al. (2020)). Denote $\{\boldsymbol{x}^{(i)}\}_{i=1}^{nK} = \{\boldsymbol{x}_{t,a} \in \mathbb{R}^d : t \in [n], a \in [K]\}, \tilde{\boldsymbol{H}}_{i,j}^{(1)} = \boldsymbol{\Sigma}_{i,j}^{(1)} = \langle \boldsymbol{x}^{(i)}, \boldsymbol{x}^{(j)} \rangle$, and

$$\boldsymbol{A}_{i,j}^{(l)} = \begin{bmatrix} \boldsymbol{\Sigma}_{i,i}^{(l)} & \boldsymbol{\Sigma}_{i,j}^{(l)} \\ \boldsymbol{\Sigma}_{i,j}^{(l)} & \boldsymbol{\Sigma}_{j,j}^{(l)} \end{bmatrix}, \quad \boldsymbol{\Sigma}_{i,j}^{(l+1)} = 2\mathbb{E}_{(u,v) \sim \mathcal{N}(\boldsymbol{0}, \boldsymbol{A}_{i,j}^{(l)})}[\sigma(u)\sigma(v)],$$

$$\tilde{\boldsymbol{H}}_{i,j}^{(l+1)} = 2\tilde{\boldsymbol{H}}_{i,j}^{(l)}\mathbb{E}_{(u,v)\sim\mathcal{N}(\boldsymbol{0},\boldsymbol{A}_{i,j}^{(l)})}\left[\sigma'(u)\sigma'(v)\right] + \boldsymbol{\Sigma}_{i,j}^{(l+1)}.$$

The neural tangent kernel (NTK) matrix is then defined as $\boldsymbol{H} = (\tilde{\boldsymbol{H}}^{(L)} + \boldsymbol{\Sigma}^{(L)})/2$.

Here, the Gram matrix $\boldsymbol{H}$ is defined recursively from the first to the last layer of the neural network using Gaussian distributions for the observed contexts $\{\boldsymbol{x}^{(i)}\}_{i=1}^{nK}$. Next, we introduce the assumptions for our analysis. First, we make an assumption about the NTK matrix $\boldsymbol{H}$ and the input data.

**Assumption 4.1.** $\exists\lambda_0 > 0, \boldsymbol{H} \succeq \lambda_0\boldsymbol{I}$, and $\forall i \in [nK], \|\boldsymbol{x}^{(i)}\|_2 = 1$. Moreover, $[\boldsymbol{x}^{(i)}]_j = [\boldsymbol{x}^{(i)}]_{j+d/2}, \forall i \in [nK], j \in [d/2]$.

The first part of Assumption 4.1 assures that $\boldsymbol{H}$ is non-singular and that the input data lies in the unit sphere $\mathbb{S}^{d-1}$. Such assumption is commonly made in overparameterized neural network literature (Arora et al., 2019; Du et al., 2019b;a; Cao & Gu, 2019). The non-singularity is satisfied when e.g. any two contexts in $\{\boldsymbol{x}^{(i)}\}$ are not parallel (Zhou et al., 2020), and our analysis holds regardless of whether $\lambda_0$ depends on $n$. The unit sphere condition is merely for the sake of analysis and can be relaxed to the case that the input data is bounded in 2-norm. As for any input data point $\boldsymbol{x}$ such that $\|\boldsymbol{x}\|_2 = 1$ we can always construct a new input $\boldsymbol{x}' = \frac{1}{\sqrt{2}}[\boldsymbol{x}, \boldsymbol{x}]^T$, the second part of Assumption 4.1 is mild and used merely for the theoretical analysis (Zhou et al., 2020). In particular, under Assumption 4.1 and the initialization scheme in Algorithm 1, we have $f_{\boldsymbol{W}^{(0)}}(\boldsymbol{x}^{(i)}) = 0, \forall i \in [nK]$.

Next, we make an assumption on the data generation.

**Assumption 4.2.** $\forall t, \boldsymbol{x}_t$ is independent of $\mathcal{D}_{t-1}$, and $\exists\kappa \in (0, \infty), \left\|\frac{\pi^*(\cdot|\boldsymbol{x}_t)}{\mu(\cdot|\mathcal{D}_{t-1},\boldsymbol{x}_t)}\right\|_\infty \leq \kappa, \forall t \in [n]$.

The first part of Assumption 4.2 says that the full contexts are generated by a process independent of any policy. This is minimal and standard in stochastic contextual bandits (Lattimore & Szepesvári, 2020; Rashidinejad et al., 2021; Papini et al., 2021), e.g., when $\{\boldsymbol{x}_t\}_{t=1}^n \overset{i.i.d.}{\sim} \rho$. The second part of Assumption 4.2, namely empirical single-policy concentration (eSPC) condition, requires that the behaviour policy $\mu$ has sufficient coverage over only the optimal policy $\pi^*$ in the observed contexts. Our data coverage condition is significantly milder than the common uniform data coverage assumptions in the OPL literature (Munos & Szepesvári, 2008; Chen & Jiang, 2019; Brandfonbrener et al., 2021; Jin et al., 2020; Nguyen-Tang et al., 2021) that requires the offline data to be sufficiently explorative in all contexts and all actions. Moreover, our data coverage condition can be considered as an extension of the single-policy concentration condition in (Rashidinejad et al., 2021) where both require coverage over the optimal policy. However, the remarkable difference is that, unlike (Rashidinejad et al., 2021), the behaviour policy $\mu$ in our condition needs not to be stationary and the concentration is only defined on the observed contexts; that is, $a_t$ can be dependent on both $\boldsymbol{x}_t$ and $\mathcal{D}_{t-1}$. This is more practical as it is natural that the offline data was collected by an active learner such as a Q-learning agent (Mnih et al., 2015).

Next, we define the *effective dimension* of the NTK matrix on the observed data as $\tilde{d} = \frac{\log\det(\boldsymbol{I}+\boldsymbol{H}/\lambda)}{\log(1+nK/\lambda)}$. This notion of effective dimension was used in (Zhou et al., 2020) for online neural contextual bandits while a similar notion was introduced in (Valko et al., 2013) for online kernelized contextual bandits, and was also used in (Yang & Wang, 2020; Yang et al., 2020) for online kernelized reinforcement learning. Although being in offline policy learning setting, the online-like nature of NeuraLCB allows us to leverage the usefulness of the effective dimension. Intuitively, $\tilde{d}$ measures how quickly the eigenvalues of $\boldsymbol{H}$ decays. For example, $\tilde{d}$ only depends on $n$ logarithmically when the eigenvalues of $\boldsymbol{H}$ have a finite spectrum (in this case $\tilde{d}$ is smaller than the number of spectrum which is the dimension of the feature space) or are exponentially decaying (Yang et al., 2020). We are now ready to present the main result about the sub-optimality bound of NeuraLCB.

**Theorem 4.1.** *For any $\delta \in (0, 1)$, under Assumption 4.1 and 4.2, if the network width $m$, the regularization parameter $\lambda$, the confidence parameters $\{\beta_t\}$ and the learning rates $\{\eta_t\}$ in Algorithm 1 satisfy*

$$m \geq poly(n, L, K, \lambda^{-1}, \lambda_0^{-1}, \log(1/\delta)), \quad \lambda \geq \max\{1, \Theta(L)\},$$

$$\beta_t = \sqrt{\lambda + C_3^2 tL} \cdot (t^{1/2}\lambda^{-1/2} + (nK)^{1/2}\lambda_0^{-1/2}) \cdot m^{-1/2} \text{ for some absolute constant } C_3,$$

$$\eta_t = \frac{\iota}{\sqrt{t}}, \text{ where } \iota^{-1} = \Omega(n^{2/3}m^{5/6}\lambda^{-1/6}L^{17/6}\log^{1/2} m) \vee \Omega(m\lambda^{1/2}\log^{1/2}(nKL^2(10n+4)/\delta)),$$

*then with probability at least $1 - \delta$ over the randomness of $\boldsymbol{W}^{(0)}$ and $\mathcal{D}_n$, the sub-optimality of $\hat{\pi}$ returned by Algorithm 1 is bounded as*

$$n \cdot \mathbb{E}\left[\text{SubOpt}(\hat{\pi})\right] \leq \kappa\sqrt{n}\sqrt{\tilde{d}\log(1 + nK/\lambda) + 2} + \kappa\sqrt{n} + 2 + \sqrt{2n\log((10n + 4)/\delta)},$$

*where $\tilde{d}$ is the effective dimension of the NTK matrix, and $\kappa$ is the empirical single-policy concentration (eSPC) coefficient in Assumption 4.2.*

Our bound can be further simplified as $\mathbb{E}[\text{SubOpt}(\hat{\pi})] = \tilde{\mathcal{O}}(\kappa \cdot \max\{\sqrt{\tilde{d}}, 1\} \cdot n^{-1/2})$. A detailed proof for Theorem 4.1 is omitted to Section A. We make several notable remarks about our result. *First*, our bound does not scale linearly with $p$ or $\sqrt{p}$ as it would if the classical analyses (Abbasi-Yadkori et al., 2011; Jin et al., 2020) had been applied. Such a classical bound is vacuous for overparameterized neural networks where $p$ is significantly larger than $n$. Specifically, the online-like nature of NeuraLCB allows us to leverage a matrix determinant lemma and the notion of effective dimension in online learning (Abbasi-Yadkori et al., 2011; Zhou et al., 2020) which avoids the dependence on the dimension $p$ of the feature space as in the existing OPL methods such as (Jin et al., 2020). *Second*, as our bound scales linearly with $\sqrt{\tilde{d}}$ where $\tilde{d}$ scales only logarithmically with $n$ in common cases (Yang et al., 2020), our bound is sublinear in such cases and presents a provably efficient generalization. *Third*, our bound scales linearly with $\kappa$ which does not depend on the coverage of the offline data on other actions rather than the optimal ones. This eliminates the need for a strong uniform data coverage assumption that is commonly used in the offline policy learning literature (Munos & Szepesvári, 2008; Chen & Jiang, 2019; Brandfonbrener et al., 2021; Nguyen-Tang et al., 2021). Moreover, the online-like nature of our algorithm does not necessitate the stationarity of the offline policy, allowing an offline policy with correlated structures as in many practical scenarios. Note that Zhan et al. (2021) have also recently addressed the problem of off-policy evaluation with such offline adaptive data but used doubly robust estimators instead of direct methods as in our paper. *Fourth*, compared to the regret bound for online learning setting in (Zhou et al., 2020), we achieve an improvement by a factor of $\sqrt{\tilde{d}}$ while reducing the computational complexity from $\mathcal{O}(n^2)$ to $\mathcal{O}(n)$. On a more technical note, a key idea to achieve such an improvement is to directly regress toward the optimal parameter of the neural network instead of toward the empirical risk minimizer as in (Zhou et al., 2020).

*Finally*, to further emphasize the significance of our theoretical result, we summarize and compare it with the state-of-the-art (SOTA) sub-optimality bounds for OPL with function approximation in Table 1. From the leftmost to the rightmost column, the table describes: the related works – the function approximation – the types of algorithms where *Pessimism* means a pessimism principle based on a lower confidence bound of the reward function while *Greedy* indicates being uncertainty-agnostic (i.e., an algorithm takes an action with the highest predicted score in a given context) – the optimization problems in OPL where *Analytical* means optimization has an analytical solution, *Oracle* means the algorithm relies on an oracle to obtain the global minimizer, and *SGD* means the optimization is solved by stochastic gradient descent – the sub-optimality bounds – the data coverage assumptions where *Uniform* indicates sufficiently explorative data over the context and action spaces, *SPC* is the single-policy concentration condition, and *eSPC* is the empirical SPC – the nature of data generation required for the respective guarantees where *I* (Independent) means that the offline actions must be sampled independently while *D* (Dependent) indicates that the offline actions can be dependent on the past data. It can be seen that our result has a stronger generalization under the most practical settings as compared to the existing SOTA generalization theory for OPL. We also remark that the optimization design and guarantee in NeuraLCB (single data point SGD) are of independent interest that do not only apply to the offline setting but also to the original online setting in (Zhou et al., 2020) to improve their regret and optimization complexity.

## 5  RELATED WORK

**OPL with function approximation**. Most OPL works, in both bandit and reinforcement learning settings, use tabular representation (Yin & Wang, 2020; Buckman et al., 2020; Yin et al., 2021; Yin & Wang, 2021; Rashidinejad et al., 2021; Xiao et al., 2021) and linear models (Duan & Wang,

Table 1: The SOTA generalization theory of OPL with function approximation.

| Work | Function | Type | Optimization | Sub-optimality | Data Coverage | Data Gen. |
|------|----------|------|--------------|----------------|---------------|-----------|
| Yin & Wang (2020)[a] | Tabular | Greedy | Analytical | $\tilde{\mathcal{O}}\left(\sqrt{\|\mathcal{X}\| \cdot K} \cdot n^{-1/2}\right)$ | Uniform | I |
| Rashidinejad et al. (2021) | Tabular | Pessimism | Analytical | $\tilde{\mathcal{O}}\left(\sqrt{\|\mathcal{X}\| \cdot \kappa} \cdot n^{-1/2}\right)$ | SPC | I |
| Duan & Wang (2020)[b] | Linear | Greedy | Analytical | $\tilde{\mathcal{O}}\left(\kappa \cdot n^{-1/2} + d \cdot n^{-1}\right)$ | Uniform | I |
| Jin et al. (2020) | Linear | Pessimism | Analytical | $\tilde{\mathcal{O}}\left(d \cdot n^{-1/2}\right)$ | Uniform | I |
| Nguyen-Tang et al. (2021) | Narrow ReLU | Greedy | Oracle | $\tilde{\mathcal{O}}\left(\sqrt{\kappa} \cdot n^{-\frac{\alpha}{2(\alpha+d)}}\right)$ | Uniform | I |
| **This work** | Wide ReLU | Pessimism | SGD | $\tilde{\mathcal{O}}(\kappa \cdot \sqrt{\tilde{d}} \cdot n^{-1/2})$ | eSPC | I/D |

[a,b] The bounds of these works are for off-policy evaluation which is generally easier than OPL problem.

2020; Jin et al., 2020; Tran-The et al., 2021). The most related work on OPL with neural function approximation we are aware of are (Brandfonbrener et al., 2021; Nguyen-Tang et al., 2021; Uehara et al., 2021). However, Brandfonbrener et al. (2021); Nguyen-Tang et al. (2021); Uehara et al. (2021) rely on a strong uniform data coverage assumption and an optimization oracle to the empirical risk minimizer. Other OPL analyses with general function approximation (Duan et al., 2021; Hu et al., 2021) also use such optimization oracle, which limit the applicability of their algorithms and analyses in practical settings.

To deal with nonlinear rewards without making strong functional assumptions, other approaches rather than neural networks have been considered, including a family of experts (Auer, 2003), a reduction to supervised learning (Langford & Zhang, 2007; Agarwal et al., 2014), and nonparametric models (Kleinberg et al., 2008; Srinivas et al., 2009; Krause & Ong, 2011; Bubeck et al., 2011; Valko et al., 2013). However, they are all in the online policy learning setting instead of the OPL setting. Moreover, these approaches have time complexity scaled linearly with the number of experts, rely the regret on an oracle, and have cubic computational complexity, respectively, while our method with neural networks are both statically and computationally efficient where it achieves a $\sqrt{n}$-type suboptimality bound and only linear computational complexity.

**Neural networks**. Our work is inspired by the theoretical advances of neural networks and their subsequent application in online policy learning (Yang et al., 2020; Zhou et al., 2020; Xu et al., 2020). For optimization aspect, (stochastic) gradient descents can provably find global minima of training loss of neural networks (Du et al., 2019a;b; Allen-Zhu et al., 2019; Nguyen, 2021). For generalization aspect, (stochastic) gradient descents can train an overparameterized neural network to a regime where the neural network interpolates the training data (i.e., zero training error) and has a good generalization ability (Arora et al., 2019; Cao & Gu, 2019; Belkin, 2021).

Regarding the use of neural networks for policy learning, NeuraLCB is similar to the NeuralUCB algorithm (Zhou et al., 2020), that is proposed for the setting of online contextual bandits with neural networks, in the sense that both algorithms use neural networks, learn with a streaming data and construct a (lower and upper, respectively) confidence bound of the reward function to guide decision-making. Besides the apparent difference of offline and online policy learning problems, the notable difference of NeuraLCB from NeuralUCB is that while NeuralUCB trains a new neural network from scratch at each iteration (for multiple epochs), NeuraLCB trains a single neural network completely online. That is, NeuraLCB updates the neural network in light of the data at a current iteration from the trained network of the previous iteration. Such optimization scheme in NeuraLCB greatly reduces the computational complexity from $\mathcal{O}(n^2)$ [1] to $O(n)$, while still guaranteeing a provably efficient algorithm. Moreover, NeuraLCB achieves a suboptimality bound with a better dependence on the effective dimension than NeuralUCB.

## 6 EXPERIMENTS

In this section, we evaluate NeuraLCB and compare it with five representative baselines: (1) LinLCB (Jin et al., 2020), which also uses LCB but relies on linear models, (2) KernLCB, which approximates the reward using functions in a RKHS and is an offline counterpart of KernelUCB (Valko et al., 2013), (3) NeuralLinLCB, which is the same as LinLCB except that it uses

---

[1] In practice, at each time step $t$, NeuralUCB trains a neural network for $t$ epochs using gradient descent in the entire data collected up to time $t$.

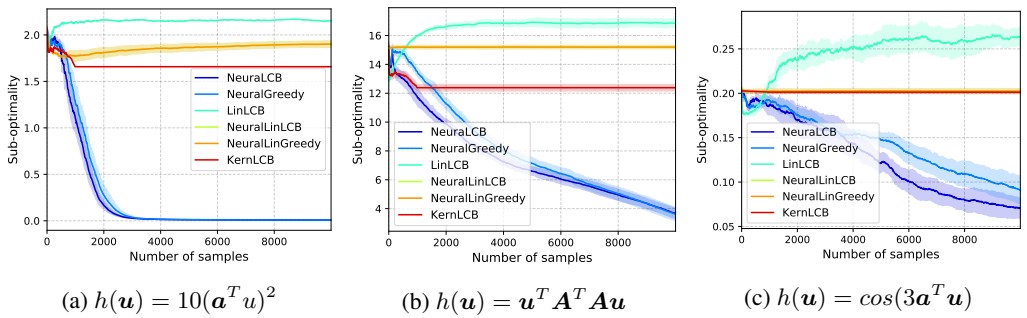

Figure 1: The sub-optimality of NeuraLCB versus the baseline algorithms on synthetic datasets.

$\phi(\boldsymbol{x}_a) = \text{vec}(\nabla f_{\boldsymbol{W}^{(0)}}(\boldsymbol{x}_a))$ as the feature extractor for the linear model where $f_{\boldsymbol{W}^{(0)}}$ is the same neural network of NeuraLCB at initialization, (4) NeuralLinGreedy, which is the same as NeuralLinLCB except that it relies on the empirical estimate of the reward function for decision-making, and (5) NeuralGreedy, which is the same as NeuraLCB except that NeuralGreedy makes decision based on the empirical estimate of the reward. For more details and completeness, we present the pseudo-code of these baseline methods in Section D. Here, we compare the algorithms by their generalization ability via their expected sub-optimality. For each algorithm, we vary the number of (offline) samples $n$ from $1$ to $T$ where $T$ will be specified in each dataset, and repeat each experiment for 10 times. We report the mean results and their $95\%$ confidence intervals. [2]

**Approximation**. To accelerate computation, we follow (Riquelme et al., 2018; Zhou et al., 2020) to approximate large covariance matrices and expensive kernel methods. Specifically, as NeuraLCB and NeuralLinLCB involve computing a covariance matrix $\boldsymbol{\Lambda}_t$ of size $p \times p$ where $p = md + m + m^2(L-2)$ is the number of the neural network parameters which could be large, we approximate $\boldsymbol{\Lambda}_t$ by its diagonal. Moreover, as KernLCB scales cubically with the number of samples, we use KernLCB fitted on the first $1,000$ samples if the offline data exceeds $1,000$ samples.

**Data generation**. We generate offline actions using a fixed $\epsilon$-greedy policy with respect to the true reward function of each considered contextual bandit, where $\epsilon$ is set to $0.1$ for all experiments in this section. In each run, we randomly sample $n_{te} = 10,000$ contexts from $\rho$ and use this same test contexts to approximate the expected sub-optimality of each algorithm.

**Hyperparameters**. We fix $\lambda = 0.1$ for all algorithms. For NeuraLCB, we set $\beta_t = \beta$, and for NeuraLCB, LinLCB, KernLCB, and NeuralLinLCB, we do grid search over $\{0.01, 0.05, 0.1, 1, 5, 10\}$ for the uncertainty parameter $\beta$. For KernLCB, we use the radius basis function (RBF) kernel with parameter $\sigma$ and do grid search over $\{0.1, 1, 10\}$ for $\sigma$. For NeuraLCB and NeuralGreedy, we use Adam optimizer (Kingma & Ba, 2014) with learning rate $\eta$ grid-searched over $\{0.0001, 0.001\}$ and set the $l_2$-regularized parameter to $0.0001$. For NeuraLCB, for each $\mathcal{D}_t$, we use $\hat{\pi}_t$ as its final returned policy instead of averaging over all policies $\{\hat{\pi}_\tau\}_{\tau=1}^t$. Moreover, we grid search NeuraLCB and NeuralGreedy over two training modes, namely $\{$S-mode, B-mode$\}$ where at each iteration $t$, S-mode updates the neural network for one step of SGD (one step of Adam update in practice) on one single data point $(\boldsymbol{x}_t, a_t, r_t)$ while B-mode updates the network for $100$ steps of SGD on a random batch of size $50$ of data $\mathcal{D}_t$ (details at Algorithm 7). We remark that even in the B-mode, NeuraLCB is still more computationally efficient than its online counterpart NeuralUCB (Zhou et al., 2020) as NeuraLCB reuses the neural network parameters from the previous iteration instead of training it from scratch for each new iteration. For NeuralLinLCB, NeuralLinGreedy, NeuraLCB, and NeuralGreedy, we use the same network architecture with $L = 2$ and add layer normalization (Ba et al., 2016) in the hidden layers. The network width $m$ will be specified later based on datasets.

## 6.1 SYNTHETIC DATASETS

For synthetic experiments, we evaluate the algorithms on contextual bandits with the synthetic non-linear reward functions $h$ used in (Zhou et al., 2020):

$$h_1(\boldsymbol{u}) = 10(\boldsymbol{u}^T\boldsymbol{a})^2, \quad h_2(\boldsymbol{u}) = \boldsymbol{u}^T\boldsymbol{A}^T\boldsymbol{A}\boldsymbol{u}, \quad h_3(\boldsymbol{u}) = \cos(3\boldsymbol{u}^T\boldsymbol{a}),$$

---

[2]Our code repos: https://github.com/thanhnguyentang/offline_neural_bandits.

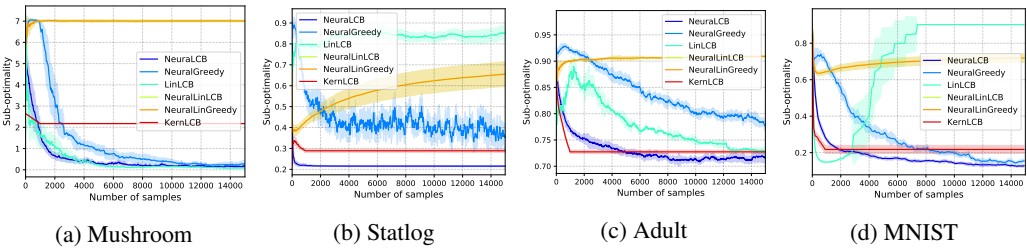

Figure 2: The sub-optimality of NeuraLCB versus the baseline algorithms on real-world datasets.

where $\boldsymbol{a} \in \mathbb{R}^d$ is randomly generated from uniform distribution over the unit sphere, and each entry of $\boldsymbol{A} \in \mathbb{R}^{d \times d}$ is independently and randomly generated from $\mathcal{N}(0, 1)$. For each reward function $h_i$, $r_t = h_i(\boldsymbol{x}_{t,a_t}) + \xi_t$ where $\xi_t \sim \mathcal{N}(0, 0.1)$. The context distribution $\rho$ for three cases is the uniform distribution over the unit sphere. All contextual bandit instances have context dimension $d = 20$ and $K = 30$ actions. Moreover, we choose the network width $m = 20$ and the maximum number of samples $T = 10,000$ for the synthetic datasets.

## 6.2 REAL-WORLD DATASETS

We evaluate the algorithms on real-world datasets from UCI Machine Learning Repository (Dua & Graff, 2017): *Mushroom*, *Statlog*, and *Adult*, and *MNIST* (LeCun et al., 1998). They represent a good range of properties: small versus large sizes, dominating actions, and stochastic versus deterministic rewards (see Section E in the appendix for details on each dataset). Besides the *Mushroom* bandit, *Statlog*, *Adult*, and *MNIST* are $K$-class classification datasets, which we convert into $K$-armed contextual bandit problems, following (Riquelme et al., 2018; Zhou et al., 2020). Specifically, for each input $\boldsymbol{x} \in \mathbb{R}^d$ in a $K$-class classification problem, we create $K$ contextual vectors $\boldsymbol{x}^1 = (\boldsymbol{x}, \boldsymbol{0}, \ldots, \boldsymbol{0}), \ldots, \boldsymbol{x}^K = (\boldsymbol{0}, \ldots, \boldsymbol{0}, \boldsymbol{x}) \in \mathbb{R}^{dK}$. The learner receives reward 1 if it selects context $\boldsymbol{x}^y$ where $y$ is the label of $\boldsymbol{x}$, and receives reward 0 otherwise. Moreover, we choose the network width $m = 100$ and the maximum sample number $T = 15,000$ for these datasets.

## 6.3 RESULTS

Figure 1 and 2 show the expected sub-optimality of all algorithms on synthetic datasets and real-world datasets, respectively. First, due to the non-linearity of the reward functions, methods with linear models (LinLCB, NeuralLinLCB, NeuralLinGreedy, and KernLCB) fail in almost all tasks (except that LinLCB and KernLCB have competitive performance in *Mushroom* and *Adult*, respectively). In particular, linear models using neural network features without training (NeuralLinLCB and NeuralLinGreedy) barely work in any datasets considered here. In contrast, our method NeuraLCB outperforms all the baseline methods in all tasks. We remark that NeuralGreedy has a substantially lower sub-optimality in synthetic datasets (and even a comparable performance with NeuraLCB in $h_1$) than linear models, suggesting the importance of using trainable neural representation in highly non-linear rewards instead of using linear models with fixed feature extractors. On the other hand, our method NeuraLCB outperforms NeuralGreedy in real-world datasets by a large margin (even though two methods are trained exactly the same but different only in decision-making), confirming the effectiveness of pessimism principle in these tasks. Second, KernLCB performs reasonably in certain OPL tasks (*Adult* and slightly well in *Statlog* and *MNIST*), but the cubic computational complexity of kernel methods make it less appealing in OPL with offline data at more than moderate sizes. Note that due to such cubic computational complexity, we follow (Riquelme et al., 2018; Zhou et al., 2020) to learn the kernel in KernLCB for only the first $1,000$ samples and keep the fitted kernel for the rest of the data (which explains the straight line of KernLCB after $n = 1,000$ in our experiments). Our method NeuraLCB, on the other hand, is highly computationally efficient as the computation scales only linearly with $n$ (even in the B-mode – a slight modification of the original algorithm to include batch training). In fact, in the real-world datasets above, the S-mode (which trains in a single data point for one SGD step at each iteration) outperforms the B-mode, further confirming the effectiveness of the online-like nature in NeuraLCB. In Section F, we reported the performance of S-mode and B-mode together and evaluated on dependent offline data.

## 7 ACKNOWLEDGEMENT

This research was partially funded by the Australian Government through the Australian Research Council (ARC). Prof. Venkatesh is the recipient of an ARC Australian Laureate Fellowship (FL170100006).

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

# A    PROOF OF THEOREM 4.1

In this section, we provide the proof of Theorem 4.1.

Let $\mathcal{D}_t = \{(\boldsymbol{x}_\tau, a_\tau, r_\tau)\}_{1 \leq \tau \leq t}$. Note that $\hat{\pi}_t$ returned by Algorithm 1 is $\mathcal{D}_{t-1}$-measurable. Denote $\mathbb{E}_t[\cdot] = \mathbb{E}[\cdot | \mathcal{D}_{t-1}, \boldsymbol{x}_t]$. Let the step sizes $\{\eta_t\}$ defined as in Theorem 4.1 and the confidence trade-off parameters $\{\beta_t\}$ defined as in Algorithm 1, we present main lemmas below which will culminate into the proof of the main theorem.

**Lemma A.1.** *There exists absolute constants $C_1, C_2 > 0$ such that for any $\delta \in (0,1)$, if $m$ satisfies*

$$m \geq \max\left\{\Theta(n\lambda^{-1}L^{11}\log^6 m), \Theta(L^6 n^4 K^4 \lambda_0^{-4} \log(KL(5n+1)/\delta))\right.$$
$$\left.\Theta(L^{-1}\lambda^{1/2}(\log^{3/2}(nKL^2(5n+1)/\delta) \vee \log^{-3/2} m))\right\},$$

*then with probability at least $1 - \delta$, it holds uniformly over all $t \in [n]$ that*

$$\mathrm{SubOpt}(\hat{\pi}_t; \boldsymbol{x}_t) \leq 2\beta_{t-1}\mathbb{E}_{a_t^* \sim \pi^*(\cdot|\boldsymbol{x}_t)}\left[\|\nabla f_{\boldsymbol{W}^{(t-1)}}(\boldsymbol{x}_{t,a_t^*}) \cdot m^{-1/2}\|_{\boldsymbol{\Lambda}_{t-1}^{-1}} | \mathcal{D}_{t-1}, \boldsymbol{x}_t\right]$$
$$+ 2C_1 t^{2/3} m^{-1/6} \log^{1/2} m L^{7/3} \lambda^{-1/2} + 2\sqrt{2}C_2\sqrt{nK}\lambda_0^{-1/2} m^{-11/6} \log^{1/2} mt^{1/6} L^{10/3} \lambda^{-1/6}.$$

Lemma A.1 gives an upper bound on the sub-optimality of the returned step-dependent policy $\hat{\pi}_t$ on the observed context $\boldsymbol{x}_t$ for each $t \in [n]$. We remark that the upper bound depends on the rate at which the confidence width of the NTK feature vectors shrink along the direction of only the optimal actions, rather than any other actions. This is an advantage of pessimism where it does not require the offline data to be informative about any sub-optimal actions. However, the upper bound depends on the unknown optimal policy $\pi^*$ while the offline data has been generated a priori by a different unknown behaviour policy. This distribution mismatch is handled in the next lemma.

**Lemma A.2.** *There exists an absolute constant $C_3 > 0$ such that for any $\delta \in (0,1)$, if $m$ and $\lambda$ satisfy*

$$\lambda \geq \max\{1, \Theta(L)\}, \quad m \geq \max\left\{\Theta(n\lambda^{-1}L^{11}\log^6 m), \Theta(L^6 n^4 K^4 \lambda_0^{-4} \log(nKL(5n+2)/\delta))\right.$$
$$\left.\Theta(L^{-1}\lambda^{1/2}(\log^{3/2}(nKL^2(5n+2)/\delta) \vee \log^{-3/2} m))\right\},$$

*then with probability at least $1 - \delta$, we have*

$$\frac{1}{n}\sum_{t=1}^n \beta_{t-1}\mathbb{E}_{a_t^* \sim \pi^*(\cdot|\boldsymbol{x}_t)}\left[\|\nabla f_{\boldsymbol{W}^{(t-1)}}(\boldsymbol{x}_{t,a_t^*}) \cdot m^{-1/2}\|_{\boldsymbol{\Lambda}_{t-1}^{-1}} | \mathcal{D}_{t-1}, \boldsymbol{x}_t\right]$$
$$\leq \frac{\sqrt{2}\beta_n \kappa}{\sqrt{n}}\sqrt{\tilde{d}\log(1 + nK/\lambda) + 1 + 2C_2 C_3^2 n^{3/2} m^{-1/6}(\log m)^{1/2}L^{23/6}\lambda^{-1/6}}$$
$$+ \beta_n \kappa(C_3/\sqrt{2})L^{1/2}\lambda_0^{-1/2}\log^{1/2}((5n+2)/\delta),$$

*where $C_2$ is from Lemma A.1.*

We also remark that the upper bound in Lemma A.2 scales linearly with $\sqrt{\tilde{d}}$ instead of with $\sqrt{p}$ if a standard analysis were applied. This avoids a vacuous bound as $p$ is large with respect to $n$.

The upper bounds in Lemma A.1 and Lemma A.2 are established for the observed contexts only. The next lemma generalizes these bounds to the entire context distribution, thanks to the online-like nature of Algorithm 1 and an online-to-batch argument. In particular, a key technical property of Algorithm 1 that makes this generalization possible without a uniform convergence is that $\hat{\pi}_t$ is $\mathcal{D}_{t-1}$-measurable and independent of $(\boldsymbol{x}_t, a_t, r_t)$.

**Lemma A.3.** *For any $\delta \in (0,1)$, with probability at least $1 - \delta$ over the randomness of $\mathcal{D}_n$, we have*

$$\mathbb{E}\left[\mathrm{SubOpt}(\hat{\pi})\right] \leq \frac{1}{n}\sum_{t=1}^n \mathrm{SubOpt}(\hat{\pi}_t; \boldsymbol{x}_t) + \sqrt{\frac{2}{n}\log(1/\delta)}.$$

We are now ready to prove the main theorem.

*Proof of Theorem 4.1.* Combining Lemma A.1, Lemma A.2, and Lemma A.3 via the union bound, we have

$$n \cdot \mathbb{E}[\text{SubOpt}(\hat{\pi})] \leq \kappa\sqrt{n}\Gamma_1\sqrt{\tilde{d}\log(1+nK/\lambda)+\Gamma_2}+\kappa\sqrt{n}\Gamma_3+\Gamma_4+\Gamma_5+\sqrt{2n\log((10n+4)/\delta)},$$

$$\leq \kappa\sqrt{n}\sqrt{\tilde{d}\log(1+nK/\lambda)+2}+\kappa\sqrt{n}+2+\sqrt{2n\log(10n+4)/\delta)}$$

where $m$ is chosen to be sufficiently large as a polynomial of $(n, L, K, \lambda^{-1}, \lambda_0^{-1}, \log(1/\delta))$ such that

$$\Gamma_1 := 2\sqrt{2}\sqrt{\lambda + C_3^2 nL(n^{1/2}\lambda^{1/2}+(nK)^{1/2}\lambda_0^{-1/2})} \cdot m^{-1/2} \leq 1$$

$$\Gamma_2 := 1 + 2C_2C_3^2 n^{3/2}m^{-1/6}(\log m)^{1/2}L^{23/6}\lambda^{-1/6} \leq 2$$

$$\Gamma_3 := \Gamma_1\sqrt{n}(C_3/\sqrt{2})L^{1/2}\lambda_0^{-1/2}\log^{1/2}((10n+4)/\delta) \leq 1$$

$$\Gamma_4 := 2C_1 n^{5/3}m^{-1/6}(\log m)^{1/2}L^{7/3}\lambda^{-1/2} \leq 1$$

$$\Gamma_5 := 2\sqrt{2}C_2(nK)^{1/2}\lambda_0^{-1/2}m^{-11/6}(\log m)^{1/2}n^{7/6}L^{10/3}\lambda^{-1/6} \leq 1.$$

$\square$

# B  PROOF OF LEMMAS IN SECTION A

## B.1  PROOF OF LEMMA A.1

We start with the following lemmas whose proofs are deferred to Appendix B.

**Lemma B.1.** *Let $\boldsymbol{h} = [h(\boldsymbol{x}^{(1)}),\dots,h(\boldsymbol{x}^{(nK)})]^T \in \mathbb{R}^{nK}$. There exists $\boldsymbol{W}^* \in \mathcal{W}$ such that for any $\delta \in (0,1)$, if $m \geq \Theta(L^6 n^4 K^4 \lambda_0^{-4}\log(nKL/\delta))$, with probability at least $1-\delta$ over the randomness of $\boldsymbol{W}^{(0)}$, it holds uniformly for all $i \in [nK]$ that*

$$\|\boldsymbol{W}^* - \boldsymbol{W}^{(0)}\|_F \leq \sqrt{2}m^{-1/2}\|\boldsymbol{h}\|_{\boldsymbol{H}^{-1}},$$

$$\langle \nabla f_{\boldsymbol{W}^{(0)}}(\boldsymbol{x}^{(i)}), \boldsymbol{W}^* - \boldsymbol{W}^{(0)}\rangle = h(\boldsymbol{x}^{(i)}).$$

*Remark* B.1. Lemma B.1 shows that for a sufficiently wide network, there is a linear model that uses the gradient of the neural network at initialization as a feature vector and interpolates the reward function in the training inputs. Moreover, the weights $\boldsymbol{W}^*$ of the linear model is in a neighborhood of the initialization $\boldsymbol{W}^{(0)}$. Note that we also have

$$S := \|\boldsymbol{h}\|_{\boldsymbol{H}^{-1}} \leq \|\boldsymbol{h}\|_2\sqrt{\|\boldsymbol{H}^{-1}\|_2} \leq \sqrt{nK}\lambda_0^{-1/2},$$

where the second inequality is by Assumption 4.1 and Cauchy-Schwartz inequality with $h(\boldsymbol{x}) \in [0,1], \forall \boldsymbol{x}$.

**Lemma B.2.** *For any $\delta \in (0,1)$, if $m$ satisfies*

$$m \geq \Theta(n\lambda^{-1}L^{11}\log^6 m) \vee \Theta(L^{-1}\lambda^{1/2}\log^{3/2}(3n^2KL^2/\delta)),$$

*and the step sizes satisfy*

$$\eta_t = \frac{\iota}{\sqrt{t}} \text{ where } \iota^{-1} = \Omega(n^{2/3}m^{5/6}\lambda^{-1/6}L^{17/6}\log^{1/2}m) \vee \Omega(Rm\lambda^{1/2}\log^{1/2}(n/\delta))$$

*then with probability at least $1-\delta$ over the randomness of $\boldsymbol{W}^{(0)}$ and $\mathcal{D}$, it holds uniformly for all $t \in [n], l \in [L]$ that*

$$\|\boldsymbol{W}_l^{(t)} - \boldsymbol{W}_l^{(0)}\|_F \leq \sqrt{\frac{t}{m\lambda L}}, \text{ and } \|\boldsymbol{\Lambda}_t\|_2 \leq \lambda + C_3^2 tL,$$

*where $C_3 > 0$ is an absolute constant from Lemma C.2.*

*Remark* B.2. Lemma B.2 controls the growth dynamics of the learned weights $\boldsymbol{W}_t$ around its initialization and bounds the spectral norm of the empirical covariance matrix $\boldsymbol{\Lambda}_t$ when the model is trained by SGD.

**Lemma B.3** (Allen-Zhu et al. (2019, Theorem 5), Cao & Gu (2019, Lemma B.5)). *There exist an absolute constant $C_2 > 0$ such that for any $\delta \in (0,1)$, if $\omega$ satisfies*

$$\Theta(m^{-3/2}L^{-3/2}(\log^{3/2}(nK/\delta)) \vee \log^{-3/2} m) \leq \omega \leq \Theta(L^{-9/2}\log^{-3} m),$$

*with probability at least $1 - \delta$ over the randomness of $\boldsymbol{W}^{(0)}$, it holds uniformly for all $\boldsymbol{W} \in \mathcal{B}(\boldsymbol{W}^{(0)}; \omega)$ and $i \in [nK]$ that*

$$\|\nabla f_{\boldsymbol{W}}(\boldsymbol{x}^{(i)}) - \nabla f_{\boldsymbol{W}^{(0)}}(\boldsymbol{x}^{(i)})\|_F \leq C_2\sqrt{\log m}\,\omega^{1/3}L^3\|\nabla f_{\boldsymbol{W}^{(0)}}(\boldsymbol{x}^{(i)})\|_F.$$

*Remark* B.3. Lemma B.3 shows that the gradient in a neighborhood of the initialization differs from the gradient at the initialization by an amount that can be explicitly controlled by the radius of the neighborhood and the norm of the gradient at initialization.

**Lemma B.4** (Cao & Gu (2019, Lemma 4.1)). *There exist an absolute constant $C_1 > 0$ such that for any $\delta \in (0,1)$ over the randomness of $\boldsymbol{W}^{(0)}$, if $\omega$ satisfies*

$$\Theta(m^{-3/2}L^{-3/2}\log^{3/2}(nKL^2/\delta)) \leq \omega \leq \Theta(L^{-6}\log^{-3/2} m),$$

*with probability at least $1 - \delta$, it holds uniformly for all $\boldsymbol{W}, \boldsymbol{W}' \in \mathcal{B}(\boldsymbol{W}^{(0)}; \omega)$ and $i \in [nK]$ that*

$$|f_{\boldsymbol{W}'}(\boldsymbol{x}^{(i)}) - f_{\boldsymbol{W}}(\boldsymbol{x}^{(i)}) - \langle\nabla f_{\boldsymbol{W}}(\boldsymbol{x}^{(i)}), \boldsymbol{W}' - \boldsymbol{W}\rangle| \leq C_1 \cdot \omega^{4/3}L^3\sqrt{m\log m}.$$

*Remark* B.4. Lemma B.4 shows that near initialization the neural network function is almost linear in terms of its weights in the training inputs.

*Proof of Lemma A.1.* For all $t \in [n], \boldsymbol{u} \in \mathbb{R}^d$, we define

$$U_t(\boldsymbol{u}) = f_{\boldsymbol{W}^{(t-1)}}(\boldsymbol{u}) + \beta_{t-1}\|\nabla f_{\boldsymbol{W}^{(t-1)}}(\boldsymbol{u}) \cdot m^{-1/2}\|_{\boldsymbol{\Lambda}_{t-1}^{-1}}$$

$$L_t(\boldsymbol{u}) = f_{\boldsymbol{W}^{(t-1)}}(\boldsymbol{u}) - \beta_{t-1}\|\nabla f_{\boldsymbol{W}^{(t-1)}}(\boldsymbol{u}) \cdot m^{-1/2}\|_{\boldsymbol{\Lambda}_{t-1}^{-1}}$$

$$\bar{U}_t(\boldsymbol{u}) = \langle\nabla f_{\boldsymbol{W}^{(t-1)}}(\boldsymbol{u}), \boldsymbol{W}^{(t-1)} - \boldsymbol{W}^{(0)}\rangle + \beta_{t-1}\|\nabla f_{\boldsymbol{W}^{(t-1)}}(\boldsymbol{u}) \cdot m^{-1/2}\|_{\boldsymbol{\Lambda}_{t-1}^{-1}}$$

$$\bar{L}_t(\boldsymbol{u}) = \langle\nabla f_{\boldsymbol{W}^{(t-1)}}(\boldsymbol{u}), \boldsymbol{W}^{(t-1)} - \boldsymbol{W}^{(0)}\rangle - \beta_{t-1}\|\nabla f_{\boldsymbol{W}^{(t-1)}}(\boldsymbol{u}) \cdot m^{-1/2}\|_{\boldsymbol{\Lambda}_{t-1}^{-1}}$$

$$\mathcal{C}_t = \{\boldsymbol{W} \in \mathcal{W} : \|\boldsymbol{W} - \boldsymbol{W}^{(t-1)}\|_{\boldsymbol{\Lambda}_{t-1}} \leq \beta_{t-1}\}.$$

Let $\mathcal{E}$ be the event in which Lemma B.1, Lemma B.3, Lemma B.3 for all $\omega \in \left\{\sqrt{\frac{i}{m\lambda L}} : 1 \leq i \leq n\right\}$, and Lemma B.4 for all $\omega \in \left\{\sqrt{\frac{i}{m\lambda L}} : 1 \leq i \leq n\right\}$ hold simultaneously.

Under event $\mathcal{E}$, for all $t \in [n]$, we have

$$\begin{aligned}
\|\boldsymbol{W}^* - \boldsymbol{W}^{(t)}\|_{\boldsymbol{\Lambda}_t} &\leq \|\boldsymbol{W}^* - \boldsymbol{W}^{(t)}\|_F\sqrt{\|\boldsymbol{\Lambda}_t\|_2} \\
&\leq (\|\boldsymbol{W}^* - \boldsymbol{W}^{(0)}\|_F + \|\boldsymbol{W}^{(t)} - \boldsymbol{W}^{(0)}\|_F)\sqrt{\|\boldsymbol{\Lambda}_t\|_2} \\
&\leq (\sqrt{2}m^{-1/2}S + t^{1/2}\lambda^{-1/2}m^{-1/2})\sqrt{\lambda + C_3^2 tL} = \beta_t,
\end{aligned}$$

where the second inequality is by the triangle inequality, and the third inequality is by Lemma B.1 and Lemma B.2. Thus, $\boldsymbol{W}^* \in \mathcal{C}_t, \forall t \in [n]$.

Denoting $a_t^* \sim \pi^*(\cdot|\boldsymbol{x}_t)$ and $\hat{a}_t \sim \hat{\pi}_t(\cdot|\boldsymbol{x}_t)$, under event $\mathcal{E}$, we have

$$\begin{aligned}
\text{SubOpt}(\hat{\pi}_t; \boldsymbol{x}_t) &= \mathbb{E}_t[h(\boldsymbol{x}_{t,a_t^*})] - \mathbb{E}_t[h(\boldsymbol{x}_{t,\hat{a}_t})] \\
&\overset{(a)}{=} \mathbb{E}_t\left[\langle\nabla f_{\boldsymbol{W}^{(0)}}(\boldsymbol{x}_{t,a_t^*}), \boldsymbol{W}^* - \boldsymbol{W}^{(0)}\rangle\right] - \mathbb{E}_t\left[\langle\nabla f_{\boldsymbol{W}^{(0)}}(\boldsymbol{x}_{t,\hat{a}_t}), \boldsymbol{W}^* - \boldsymbol{W}^{(0)}\rangle\right] \\
&\overset{(b)}{\leq} \mathbb{E}_t\left[\langle\nabla f_{\boldsymbol{W}^{(t-1)}}(\boldsymbol{x}_{t,a_t^*}), \boldsymbol{W}^* - \boldsymbol{W}^{(0)}\rangle\right] - \mathbb{E}_t\left[\langle\nabla f_{\boldsymbol{W}^{(t-1)}}(\boldsymbol{x}_{t,\hat{a}_t}), \boldsymbol{W}^* - \boldsymbol{W}^{(0)}\rangle\right]
\end{aligned}$$

$$+ \|\boldsymbol{W}^* - \boldsymbol{W}^{(0)}\|_F \cdot \mathbb{E}_t \left[ \|\nabla f_{\boldsymbol{W}^{(t-1)}}(\boldsymbol{x}_{t,a_t^*}) - \nabla f_{\boldsymbol{W}^{(0)}}(\boldsymbol{x}_{t,a_t^*})\|_F \right.$$

$$\left. + \|\nabla f_{\boldsymbol{W}^{(t-1)}}(\boldsymbol{x}_{t,\hat{a}_t}) - \nabla f_{\boldsymbol{W}^{(0)}}(\boldsymbol{x}_{t,\hat{a}_t})\|_F \right]$$

$$\overset{(c)}{\leq} \mathbb{E}_t \left[ \langle \nabla f_{\boldsymbol{W}^{(t-1)}}(\boldsymbol{x}_{t,a_t^*}), \boldsymbol{W}^* - \boldsymbol{W}^{(0)} \rangle \right] - \mathbb{E}_t \left[ \langle \nabla f_{\boldsymbol{W}^{(t-1)}}(\boldsymbol{x}_{t,\hat{a}_t}), \boldsymbol{W}^* - \boldsymbol{W}^{(0)} \rangle \right]$$

$$+ 2\sqrt{2} C_2 S m^{-11/6} \log^{1/2} m t^{1/6} L^{10/3} \lambda^{-1/6}$$

$$\overset{(d)}{\leq} \mathbb{E}_t \left[ \bar{U}_t(\boldsymbol{x}_{t,a_t^*}) \right] - \mathbb{E}_t \left[ \bar{L}_t(\boldsymbol{x}_{t,\hat{a}_t}) \right] + 2\sqrt{2} C_2 S m^{-11/6} \log^{1/2} m t^{1/6} L^{10/3} \lambda^{-1/6}$$

$$= \mathbb{E}_t \left[ U_t(\boldsymbol{x}_{t,a_t^*}) \right] - \mathbb{E}_t \left[ L_t(\boldsymbol{x}_{t,\hat{a}_t}) \right]$$

$$+ \mathbb{E}_t \left[ \langle \nabla f_{\boldsymbol{W}^{(t-1)}}(\boldsymbol{x}_{t,a_t^*}), \boldsymbol{W}^{(t-1)} - \boldsymbol{W}^{(0)} \rangle - f_{\boldsymbol{W}^{(t-1)}}(\boldsymbol{x}_{t,a_t^*}) + f_{\boldsymbol{W}^{(0)}}(\boldsymbol{x}_{t,a_t^*}) \right]$$

$$+ \mathbb{E}_t \left[ f_{\boldsymbol{W}^{(t-1)}}(\boldsymbol{x}_{t,\hat{a}_t}) - f_{\boldsymbol{W}^{(0)}}(\boldsymbol{x}_{t,\hat{a}_t}) - \langle \nabla f_{\boldsymbol{W}^{(t-1)}}(\boldsymbol{x}_{t,\hat{a}_t}), \boldsymbol{W}^{(t-1)} - \boldsymbol{W}^{(0)} \rangle \right]$$

$$\underbrace{- \mathbb{E}_t \left[ f_{\boldsymbol{W}^{(0)}}(\boldsymbol{x}_{t,a_t^*}) \right] + \mathbb{E}_t \left[ f_{\boldsymbol{W}^{(0)}}(\boldsymbol{x}_{t,\hat{a}_t}) \right]}_{=0 \text{ by symmetry at initialization}} + 2\sqrt{2} C_2 S m^{-11/6} \log^{1/2} m t^{1/6} L^{10/3} \lambda^{-1/6}$$

$$\overset{(e)}{\leq} \mathbb{E}_t \left[ U_t(\boldsymbol{x}_{t,a_t^*}) \right] - \mathbb{E}_t \left[ L_t(\boldsymbol{x}_{t,a_t^*}) \right] + \underbrace{\left( \mathbb{E}_t \left[ L_t(\boldsymbol{x}_{t,a_t^*}) \right] - \mathbb{E}_t \left[ L_t(\boldsymbol{x}_{t,\hat{a}_t}) \right] \right)}_{\leq 0 \text{ by pessimism}}$$

$$+ 2C_1 t^{2/3} m^{-1/6} \log^{1/2} m L^{7/3} \lambda^{-1/2} + 2\sqrt{2} C_2 S m^{-11/6} \log^{1/2} m t^{1/6} L^{10/3} \lambda^{-1/6}$$

$$\overset{(f)}{\leq} \mathbb{E}_t \left[ U_t(\boldsymbol{x}_{t,a_t^*}) \right] - \mathbb{E}_t \left[ L_t(\boldsymbol{x}_{t,a_t^*}) \right]$$

$$+ 2C_1 t^{2/3} m^{-1/6} \log^{1/2} m L^{7/3} \lambda^{-1/2} + 2\sqrt{2} C_2 S m^{-11/6} \log^{1/2} m t^{1/6} L^{10/3} \lambda^{-1/6}$$

$$= 2\beta_{t-1} \mathbb{E}_t \left[ \|\nabla f_{\boldsymbol{W}^{(t-1)}}(\boldsymbol{x}_{t,a_t^*}) \cdot m^{-1/2}\|_{\boldsymbol{\Lambda}_{t-1}^{-1}} \right]$$

$$+ 2C_1 t^{2/3} m^{-1/6} \log^{1/2} m L^{7/3} \lambda^{-1/2} + 2\sqrt{2} C_2 S m^{-11/6} \log^{1/2} m t^{1/6} L^{10/3} \lambda^{-1/6}$$

where $(a)$ is by Lemma B.1, $(b)$ is by the triangle inequality, $(c)$ is by Lemma B.1, Lemma B.2, and Lemma B.3, $(d)$ is by $\boldsymbol{W}^* \in \mathcal{C}_t$, and by that $\max_{\boldsymbol{u}:\|\boldsymbol{u}-\boldsymbol{b}\|_A \leq \gamma} \langle \boldsymbol{a}, \boldsymbol{u} - \boldsymbol{b}_0 \rangle = \langle \boldsymbol{a}, \boldsymbol{b} - \boldsymbol{b}_0 \rangle + \gamma \|\boldsymbol{a}\|_{A^{-1}}$, and $\min_{\boldsymbol{u}:\|\boldsymbol{u}-\boldsymbol{b}\|_A \leq \gamma} \langle \boldsymbol{a}, \boldsymbol{u} - \boldsymbol{b}_0 \rangle = \langle \boldsymbol{a}, \boldsymbol{b} - \boldsymbol{b}_0 \rangle - \gamma \|\boldsymbol{a}\|_{A^{-1}}$, $(e)$ is by Lemma B.4 and by that $f_{\boldsymbol{W}^{(0)}}(\boldsymbol{x}^{(i)}) = 0, \forall i \in [nK]$, and $(f)$ is by that $\hat{a}_t$ is sampled from the policy $\hat{\pi}_t$ which is greedy with respect to $L_t$.

By the union bound and the choice of $m$, we conclude our proof. $\qquad \square$

## B.2 PROOF OF LEMMA A.2

We first present the following lemma.

**Lemma B.5.** *For any $\delta \in (0, 1)$, if $m$ satisfies*

$$m \geq \max \left\{ \Theta(n\lambda^{-1} L^{11} \log^6 m), \Theta(L^{-1}\lambda^{1/2}(\log^{3/2}(nKL^2(n+2)/\delta) \vee \log^{-3/2} m)), \right.$$

$$\left. \Theta(L^6 (nK)^4 \log(L(n+2)/\delta)) \right\},$$

*and $\lambda \geq \max\{C_3^2 L, 1\}$, then with probability at least $1 - \delta$, it holds simultaneously that*

$$\sum_{i=1}^t \|\nabla f_{\boldsymbol{W}^{(i-1)}}(\boldsymbol{x}_{i,a_i}) \cdot m^{-1/2}\|_{\boldsymbol{\Lambda}_{i-1}^{-1}}^2 \leq 2 \log \frac{\det(\boldsymbol{\Lambda}_t)}{\det(\lambda \boldsymbol{I})}, \forall t \in [n],$$

$$\left| \log \frac{\det(\boldsymbol{\Lambda}_t)}{\det(\lambda \boldsymbol{I})} - \log \frac{\det(\bar{\boldsymbol{\Lambda}}_t)}{\det(\lambda \boldsymbol{I})} \right| \leq 2C_2 C_3^2 t^{3/2} m^{-1/6} (\log m)^{1/2} L^{23/6} \lambda^{-1/6}, \forall t \in [n],$$

$$\log \frac{\det(\bar{\boldsymbol{\Lambda}}_n)}{\det(\lambda \boldsymbol{I})} \leq \tilde{d} \log(1 + nK/\lambda) + 1,$$

where $\bar{\boldsymbol{\Lambda}}_t := \lambda\boldsymbol{I} + \sum_{i=1}^{t} \text{vec}(\nabla f_{\boldsymbol{W}^{(0)}}(\boldsymbol{x}_{i,a_i})) \cdot \text{vec}(\nabla f_{\boldsymbol{W}^{(0)}}(\boldsymbol{x}_{i,a_i}))^T/m$, and $C_2, C_3 > 0$ are absolute constants from Lemma B.3 and Lemma B.2, respectively.

We are now ready to prove Lemma A.2.

*Proof of Lemma A.2.* First note that $\|\boldsymbol{\Lambda}_{t-1}\|_2 \geq \lambda, \forall t$. Let $\mathcal{E}$ be the event in which Lemma B.2, Lemma C.2 for all $\omega \in \left\{\sqrt{\frac{i}{m\lambda L}} : 1 \leq i \leq n\right\}$, and Lemma B.5 simultaneously hold. Thus, under event $\mathcal{E}$, we have

$$\|\nabla f_{\boldsymbol{W}^{(t-1)}}(\boldsymbol{x}_{t,a_t}) \cdot m^{-1/2}\|_{\boldsymbol{\Lambda}_{t-1}^{-1}} \leq \|\nabla f_{\boldsymbol{W}^{(t-1)}}(\boldsymbol{x}_{t,a_t}) \cdot m^{-1/2}\|_F \sqrt{\|\boldsymbol{\Lambda}_{t-1}^{-1}\|_2} \leq C_3 L^{1/2}\lambda^{-1/2},$$

where the second inequality is by Lemma B.2 and Lemma C.2.

Thus, by Assumption 4.2, Hoeffding's inequality, and the union bound, with probability at least $1 - \delta$, it holds simultaneously for all $t \in [n]$ that

$$2\beta_{t-1}\mathbb{E}_{a_t^* \sim \pi^*(\cdot|\boldsymbol{x}_t)} \left[\|\nabla f_{\boldsymbol{W}^{(t-1)}}(\boldsymbol{x}_{t,a_t^*}) \cdot m^{-1/2}\|_{\boldsymbol{\Lambda}_{t-1}^{-1}} | \mathcal{D}_{t-1}, \boldsymbol{x}_t\right]$$

$$\leq 2\beta_{t-1}\kappa\mathbb{E}_{a \sim \mu(\cdot|\mathcal{D}_{t-1},\boldsymbol{x}_t)} \left[\|\nabla f_{\boldsymbol{W}^{(t-1)}}(\boldsymbol{x}_{t,a}) \cdot m^{-1/2}\|_{\boldsymbol{\Lambda}_{t-1}^{-1}} | \mathcal{D}_{t-1}, \boldsymbol{x}_t\right]$$

$$\leq 2\beta_{t-1}\kappa\|\nabla f_{\boldsymbol{W}^{(t-1)}}(\boldsymbol{x}_{t,a_t}) \cdot m^{-1/2}\|_{\boldsymbol{\Lambda}_{t-1}^{-1}} + \beta_{t-1}\kappa\sqrt{2}C_3 L^{1/2}\lambda^{-1/2}\log^{1/2}((5n+2)/\delta).$$

Hence, for the choice of $m$ in Lemma A.2, with probability at least $1 - \delta$, we have

$$\frac{1}{n}\sum_{t=1}^{n}\beta_{t-1}\mathbb{E}_{a_t^* \sim \pi^*(\cdot|\boldsymbol{x}_t)} \left[\|\nabla f_{\boldsymbol{W}^{(t-1)}}(\boldsymbol{x}_{t,a_t^*}) \cdot m^{-1/2}\|_{\boldsymbol{\Lambda}_{t-1}^{-1}} | \mathcal{D}_{t-1}, \boldsymbol{x}_t\right]$$

$$\leq \frac{\beta_n\kappa}{n}\sum_{t=1}^{n}\|\nabla f_{\boldsymbol{W}^{(t-1)}}(\boldsymbol{x}_{t,a_t}) \cdot m^{-1/2}\|_{\boldsymbol{\Lambda}_{t-1}^{-1}} + \frac{C_3}{\sqrt{2}}\beta_n\kappa L^{1/2}\lambda_0^{-1/2}\log^{1/2}((5n+2)/\delta)$$

$$\leq \frac{\beta_n\kappa}{n}\sqrt{n}\sqrt{\sum_{t=1}^{n}\|\nabla f_{\boldsymbol{W}^{(t-1)}}(\boldsymbol{x}_{t,a_t}) \cdot m^{-1/2}\|_{\boldsymbol{\Lambda}_{t-1}^{-1}}^2} + \frac{C_3}{\sqrt{2}}\beta_n\kappa L^{1/2}\lambda_0^{-1/2}\log^{1/2}((5n+2)/\delta)$$

$$\leq \frac{\sqrt{2}\beta_n\kappa}{\sqrt{n}}\sqrt{\log\frac{\det(\boldsymbol{\Lambda}_n)}{\det(\lambda\boldsymbol{I})}} + \frac{C_3}{\sqrt{2}}\beta_n\kappa L^{1/2}\lambda_0^{-1/2}\log^{1/2}((5n+2)/\delta)$$

$$\leq \frac{\sqrt{2}\beta_n\kappa}{\sqrt{n}}\sqrt{\log\frac{\det(\bar{\boldsymbol{\Lambda}}_n)}{\det(\lambda\boldsymbol{I})} + 2C_2C_3^2 n^{3/2}m^{-1/6}(\log m)^{1/2}L^{23/6}\lambda^{-1/6}} + \frac{C_3}{\sqrt{2}}\beta_n\kappa L^{1/2}\lambda_0^{-1/2}\log^{1/2}((5n+2)/\delta)$$

$$\leq \frac{\sqrt{2}\beta_n\kappa}{\sqrt{n}}\sqrt{\tilde{d}\log(1 + nK/\lambda) + 1 + 2C_2C_3^2 n^{3/2}m^{-1/6}(\log m)^{1/2}L^{23/6}\lambda^{-1/6}}$$

$$+ \frac{C_3}{\sqrt{2}}\beta_n\kappa L^{1/2}\lambda_0^{-1/2}\log^{1/2}((5n+2)/\delta),$$

where the first inequality is by $\beta_t \leq \beta_n, \forall t \in [n]$, the second inequality is by Cauchy-Schwartz inequality, the second inequality and the third inequality are by Lemma B.5. $\square$

### B.3 PROOF OF LEMMA A.3

*Proof of Lemma A.3.* We follow the same online-to-batch conversion argument in (Cesa-Bianchi et al., 2004). For each $t \in [n]$, define

$$Z_t = \text{SubOpt}(\hat{\pi}_t) - \text{SubOpt}(\hat{\pi}_t; \boldsymbol{x}_t).$$

Since $\hat{\pi}_t$ is $\mathcal{D}_{t-1}$-measurable and is independent of $\boldsymbol{x}_t$, and $\boldsymbol{x}_t$ are independent of $\mathcal{D}_{t-1}$ (by Assumption 4.2), we have $\mathbb{E}[Z_t|\mathcal{D}_{t-1}] = 0, \forall t \in [n]$. Note that $-1 \leq Z_t \leq 1$. Thus, by the Hoeffding-Azuma inequality, with probability at least $1 - \delta$, we have

$$\mathbb{E}[\text{SubOpt}(\hat{\pi})] = \frac{1}{n}\sum_{t=1}^{n}\text{SubOpt}(\hat{\pi}_t) = \frac{1}{n}\sum_{t=1}^{n}\text{SubOpt}(\hat{\pi}_t; \boldsymbol{x}_t) + \frac{1}{n}\sum_{t=1}^{n}Z_t$$

$$\leq \frac{1}{n}\sum_{t=1}^{n}\mathrm{SubOpt}(\hat{\pi}_t; \boldsymbol{x}_t) + \sqrt{\frac{2}{n}\log(1/\delta)}.$$

$\square$

## C  Proof of Lemmas in Section B

### C.1  Proof of Lemma B.1

We first restate the following lemma.

**Lemma C.1** (Arora et al. (2019))**.** *There exists an absolute constant $c_1 > 0$ such that for any $\epsilon > 0, \delta \in (0, 1)$, if $m \geq c_1 L^6 \epsilon^{-4}\log(L/\delta)$, for any $i, j \in [nK]$, with probability at least $1 - \delta$ over the randomness of $\boldsymbol{W}^{(0)}$, we have*

$$|\langle \nabla f_{\boldsymbol{W}^{(0)}}(\boldsymbol{x}^{(i)}), \nabla f_{\boldsymbol{W}^{(0)}}(\boldsymbol{x}^{(j)})\rangle / m - \boldsymbol{H}_{i,j}| \leq \epsilon.$$

Lemma C.1 gives an estimation error between the kernel constructed by the gradient at initialization as a feature map and the NTK kernel. Unlike Jacot et al. (2018), Lemma C.1 quantifies an exact non-asymptotic bound for $m$.

*Proof of Lemma B.1.* Let $\boldsymbol{G} = m^{-1/2} \cdot [\mathrm{vec}(\nabla f_{\boldsymbol{W}^{(0)}}(\boldsymbol{x}^{(1)})), \ldots, \mathrm{vec}(\nabla f_{\boldsymbol{W}^{(0)}}(\boldsymbol{x}^{(nK)}))] \in \mathbb{R}^{p \times nK}$. For any $\epsilon > 0, \delta \in (0, 1)$, it follows from Lemma C.1 and union bound, if $m \geq \Theta(L^6 \epsilon^{-4}\log(nKL/\delta))$, with probability at least $1 - \delta$, we have

$$\|\boldsymbol{G}^T\boldsymbol{G} - \boldsymbol{H}\|_F \leq nK\|\boldsymbol{G}^T\boldsymbol{G} - \boldsymbol{H}\|_\infty = nK\max_{i,j}|m^{-1}\langle \nabla f_{\boldsymbol{W}^{(0)}}(\boldsymbol{x}^{(i)}), \nabla f_{\boldsymbol{W}^{(0)}}(\boldsymbol{x}^{(j)})\rangle - \boldsymbol{H}_{i,j}|$$

$$\leq nK\epsilon.$$

Under the event that the inequality above holds, by setting $\epsilon = \frac{\lambda_0}{2nK}$, we have

$$\boldsymbol{H} - \boldsymbol{G}^T\boldsymbol{G} \preceq \|\boldsymbol{H} - \boldsymbol{G}^T\boldsymbol{G}\|_2 I \preceq \|\boldsymbol{H} - \boldsymbol{G}^T\boldsymbol{G}\|_F I \preceq \frac{\lambda_0}{2}I \preceq \frac{1}{2}\boldsymbol{H}. \tag{2}$$

Let $\boldsymbol{G} = \boldsymbol{P}\boldsymbol{\Lambda}\boldsymbol{Q}^T$ be the singular value decomposition of $\boldsymbol{G}$ where $\boldsymbol{P} \in \mathbb{R}^{p \times nK}, \boldsymbol{Q} \in \mathbb{R}^{nK \times nK}$ have orthogonal columns, and $\boldsymbol{\Lambda} \in \mathbb{R}^{nK \times nK}$ is a diagonal matrix. Since $\boldsymbol{G}^T\boldsymbol{G} \succeq \frac{1}{2}\boldsymbol{H} \succeq \frac{\lambda_0}{2}I$ is positive definite, $\boldsymbol{\Lambda}$ is invertible. Let $\boldsymbol{W}^* \in \mathcal{W}$ such that $\mathrm{vec}(\boldsymbol{W}^*) = \mathrm{vec}(\boldsymbol{W}^{(0)}) + m^{-1/2} \cdot \boldsymbol{P}\boldsymbol{\Lambda}^{-1}\boldsymbol{Q}^T\boldsymbol{h}$, we have

$$m^{1/2} \cdot \boldsymbol{G}^T(\mathrm{vec}(\boldsymbol{W}^*) - \mathrm{vec}(\boldsymbol{W}^{(0)})) = \boldsymbol{Q}\boldsymbol{\Lambda}\boldsymbol{P}^T\boldsymbol{P}\boldsymbol{\Lambda}^{-1}\boldsymbol{Q}^T\boldsymbol{h} = \boldsymbol{h}.$$

Moreover, we have

$$m\|\boldsymbol{W}^* - \boldsymbol{W}^{(0)}\|_F^2 = m\|\mathrm{vec}(\boldsymbol{W}^*) - \mathrm{vec}(\boldsymbol{W}^{(0)})\|_2^2 = \boldsymbol{h}^T\boldsymbol{Q}\boldsymbol{\Lambda}^{-1}\boldsymbol{P}^T\boldsymbol{P}\boldsymbol{\Lambda}^{-1}\boldsymbol{Q}^T\boldsymbol{h}$$

$$= \boldsymbol{h}^T\boldsymbol{Q}\boldsymbol{\Lambda}^{-2}\boldsymbol{Q}^T\boldsymbol{h} = \boldsymbol{h}^T(\boldsymbol{G}^T\boldsymbol{G})^{-1}\boldsymbol{h} \leq 2\boldsymbol{h}^T\boldsymbol{H}^{-1}\boldsymbol{h},$$

where the inequality is by Equation (2). $\square$

### C.2  Proof of Lemma B.2

We present the following lemma that will be used in this proof.

**Lemma C.2** (Cao & Gu (2019, Lemma B.3))**.** *There exist an absolute constant $C_3 > 0$ such that for any $\delta \in (0, 1)$ over the randomness of $\boldsymbol{W}^{(0)}$, if $\omega$ satisfies*

$$\Theta(m^{-3/2}L^{-3/2}\log^{3/2}(nKL^2/\delta)) \leq \omega \leq \Theta(L^{-6}\log^{-3} m),$$

*with probability at least $1 - \delta$, it holds uniformly for all $\boldsymbol{W} \in \mathcal{B}(\boldsymbol{W}^{(0)}; \omega), i \in [nK], l \in [L]$ that*

$$\|\nabla_l f_{\boldsymbol{W}}(\boldsymbol{x}^{(i)})\|_F \leq C_3 \cdot \sqrt{m}.$$

*Proof of Lemma B.2.* Let $\delta \in (0,1)$. Let $L_t(\boldsymbol{W}) = \frac{1}{2}(f_{\boldsymbol{W}}(\boldsymbol{x}_{t,a_t}) - r_t)^2 + \frac{m\lambda}{2}\|\boldsymbol{W} - \boldsymbol{W}^{(0)}\|_F^2$ be the regularized squared loss function on the data point $(\boldsymbol{x}_{t,a_t}, r_t)$. Recall that $\boldsymbol{W}^{(t)} = \boldsymbol{W}^{(t-1)} - \eta_t \nabla L_t(\boldsymbol{W}^{(t-1)})$. By Hoelfding's inequality and that $r_t$ is $R$-subgaussian, for any $t \in [n]$, with probability at least $1 - \delta$, we have

$$|r_t| \le |\mathbb{E}_t[r_t]| + R\sqrt{2\log(2/\delta)} = |\mathbb{E}\left[\mathbb{E}[r_t | \mathcal{D}_{t-1}, \boldsymbol{x}_t, a_t]\right]| + R\sqrt{2\log(2/\delta)}$$
$$= \mathbb{E}\left[h(\boldsymbol{x}_{t,a_t})|\mathcal{D}_{t-1}, \boldsymbol{x}_t, a_t\right] + R\sqrt{2\log(2/\delta)} \le 1 + R\sqrt{2\log(2/\delta)}. \tag{3}$$

By union bound and (3), for any sequence $\{\omega_t\}_{t\in[n]}$ such that $\Theta(m^{-3/2}L^{-3/2}\log^{3/2}(3n^2KL^2/\delta)) \le \omega_t \le \Theta(L^{-6}\log^{-3}m) \wedge \Theta(L^{-6}\log^{-3/2}m), \forall t \in [n]$, with probability at least $1 - \delta$, it holds uniformly for all $\boldsymbol{W} \in \mathcal{B}(\boldsymbol{W}^{(0)}; \omega_t), l \in [L], t \in [n]$ that

$$\|\nabla_l L_t(\boldsymbol{W})\|_F = \|\nabla_l f_{\boldsymbol{W}}(\boldsymbol{x}_{t,a_t})(f_{\boldsymbol{W}}(\boldsymbol{x}_{t,a_t}) - r_t) + m\lambda(\boldsymbol{W}_l - \boldsymbol{W}_l^{(0)})\|_F$$
$$= \|\nabla_l f_{\boldsymbol{W}}(\boldsymbol{x}_{t,a_t})(f_{\boldsymbol{W}}(\boldsymbol{x}_{t,a_t}) - f_{\boldsymbol{W}^{(0)}}(\boldsymbol{x}_{t,a_t}) - \langle \nabla f_{\boldsymbol{W}^{(0)}}(\boldsymbol{x}_{t,a_t}), \boldsymbol{W} - \boldsymbol{W}^{(0)} \rangle)$$
$$+ \nabla_l f_{\boldsymbol{W}}(\boldsymbol{x}_{t,a_t})f_{\boldsymbol{W}^{(0)}}(\boldsymbol{x}_{t,a_t}) + \nabla_l f_{\boldsymbol{W}}(\boldsymbol{x}_{t,a_t})\langle \nabla f_{\boldsymbol{W}^{(0)}}(\boldsymbol{x}_{t,a_t}), \boldsymbol{W} - \boldsymbol{W}^{(0)} \rangle$$
$$- r_t \nabla_l f_{\boldsymbol{W}}(\boldsymbol{x}_{t,a_t}) + m\lambda(\boldsymbol{W}_l - \boldsymbol{W}_l^{(0)})\|_F$$
$$\le \|\nabla_l f_{\boldsymbol{W}}(\boldsymbol{x}_{t,a_t})(f_{\boldsymbol{W}}(\boldsymbol{x}_{t,a_t}) - f_{\boldsymbol{W}^{(0)}}(\boldsymbol{x}_{t,a_t}) - \langle \nabla f_{\boldsymbol{W}^{(0)}}(\boldsymbol{x}_{t,a_t}), \boldsymbol{W} - \boldsymbol{W}^{(0)} \rangle)\|_F$$
$$+ \underbrace{\|\nabla_l f_{\boldsymbol{W}}(\boldsymbol{x}_{t,a_t})f_{\boldsymbol{W}^{(0)}}(\boldsymbol{x}_{t,a_t})\|_F}_{=0} + \|\nabla_l f_{\boldsymbol{W}}(\boldsymbol{x}_{t,a_t})\langle \nabla f_{\boldsymbol{W}^{(0)}}(\boldsymbol{x}_{t,a_t}), \boldsymbol{W} - \boldsymbol{W}^{(0)} \rangle\|_F$$
$$+ |r_t|\|\nabla_l f_{\boldsymbol{W}}(\boldsymbol{x}_{t,a_t})\|_F + m\lambda\|\boldsymbol{W}_l - \boldsymbol{W}_l^{(0)}\|_F$$
$$\le C_1 C_3 \omega_t^{4/3} L^3 m \log^{1/2} m + C_3 m^{1/2}(1 + R\sqrt{2\log(6n/\delta)}) + C_3^2 Lm\omega + m\lambda\omega_t, \tag{4}$$

where the first inequality is triangle's inequality, and the second inequality is by Lemma B.4, Lemma C.2 and (3).

We now prove by induction that under the same event that (4) with $\omega_t = \sqrt{\frac{t}{m\lambda L}}$ holds, $\boldsymbol{W}^{(t)} \in \mathcal{B}(\boldsymbol{W}^{(0)}; \sqrt{\frac{t}{m\lambda L}}), \forall t \in [n]$. It trivially holds for $t = 0$. Assume $\boldsymbol{W}^{(i)} \in \mathcal{B}(\boldsymbol{W}^{(0)}; \sqrt{\frac{i}{m\lambda L}}), \forall i \in [t-1]$, we will prove that $\boldsymbol{W}^{(t)} \in \mathcal{B}(\boldsymbol{W}^{(0)}; \sqrt{\frac{t}{m\lambda L}})$. Indeed, it is easy to verify that there exist absolute constants $\{C_i\}_{i=1}^2 > 0$ such that if $m$ satisfies the inequalities in Lemma B.2, then $\Theta(m^{-3/2}L^{-3/2}\log^{3/2}(3n^2KL^2/\delta)) \le \sqrt{\frac{i}{m\lambda L}} \le \Theta(L^{-6}\log^{-3}m) \wedge \Theta(L^{-6}\log^{-3/2}m), \forall i \in [n]$. Thus, under the same event that (4) holds, we have

$$\|\boldsymbol{W}_l^{(t)} - \boldsymbol{W}_l^{(0)}\|_F \le \sum_{i=1}^t \|\boldsymbol{W}_l^{(i)} - \boldsymbol{W}_l^{(i-1)}\|_F = \sum_{i=1}^t \eta_i \|\nabla_l L_i(\boldsymbol{W}^{(i-1)})\|_F$$
$$\le \iota t C_1 C_3 m^{1/3} \lambda^{-2/3} L^{7/3} n^{1/6} \log^{1/2} m + \iota t m^{-1/2} \lambda^{-1/2} L^{-1/2}(C_3^2 L + \lambda)$$
$$+ 2C_3 \iota \sqrt{t} m^{1/2}(1 + R\sqrt{2}\log^{1/2}(6n/\delta))$$
$$\le \frac{1}{2}\sqrt{\frac{t}{m\lambda L}} + \frac{1}{2}\sqrt{\frac{t}{m\lambda L}} = \sqrt{\frac{t}{m\lambda L}},$$

where the first inequality is by triangle's inequality, the first equation is by the SGD update for each $\boldsymbol{W}^{(i)}$, the second inequality is by (4) and the last inequality is due to $\iota$ be chosen as

$$\begin{cases} \iota^{-1} \ge 4C_3 m\lambda^{1/2}L^{1/2}\left(1 + R\sqrt{2}\log^{1/2}(6n/\delta)\right) \\ \iota^{-1} \ge 2n^{1/2}m^{1/2}\lambda^{1/2}L^{1/2}\left(C_1 C_3 m^{1/3}\lambda^{-2/3}L^{7/3}n^{1/6}\log^{1/2}m + m^{-1/2}\lambda^{-1/2}L^{-1/2}(C_3^2 L + \lambda)\right), \end{cases}$$

which is satisfied for $\iota^{-1} = \Omega(n^{2/3}m^{5/6}\lambda^{-1/6}L^{17/6}\log^{1/2}m) \vee \Omega(Rm\lambda^{1/2}\log^{1/2}(n/\delta))$.

For the second part of the lemma, we have

$$\|\boldsymbol{\Lambda}_t\|_2 = \|\lambda \boldsymbol{I} + \sum_{i=1}^t \text{vec}(\nabla f_{\boldsymbol{W}^{(i-1)}}(\boldsymbol{x}_{i,a_i})) \cdot \text{vec}(\nabla f_{\boldsymbol{W}^{(i-1)}}(\boldsymbol{x}_{i,a_i}))^T / m\|_2$$

$$\leq \lambda + \sum_{i=1}^{t} \|\nabla f_{\boldsymbol{W}^{(i-1)}}(\boldsymbol{x}_{i,a_i})\|_F^2/m \leq \lambda + C_3^2 tL,$$

where the first inequality is by triangle's inequality and the second inequality is by $\|\boldsymbol{W}^{(i)} - \boldsymbol{W}^{(0)}\|_F \leq \sqrt{\frac{i}{m\lambda}}, \forall i \in [n]$ and Lemma C.2. $\qquad\square$

### C.3   PROOF OF LEMMA B.5

*Proof of Lemma B.5.* Let $\mathcal{E}(\delta)$ be the event in which the following $(n+2)$ events hold simultaneously: the events in which Lemma B.3 for each $\omega \in \left\{\sqrt{\frac{i}{m\lambda L}} : 1 \leq i \leq n\right\}$ holds, the event in which Lemma C.1 for $\epsilon = (nK)^{-1}$ holds, and the event in which Lemma C.2 holds.
Under event $\mathcal{E}(\delta)$, we have

$$\|\nabla f_{\boldsymbol{W}^{(t-1)}}(\boldsymbol{x}_{t,a_t}) \cdot m^{-1/2}\|_F \leq C_3 L^{1/2}, \forall t \in [n].$$

Thus, by (Abbasi-Yadkori et al., 2011, Lemma 11), if we choose $\lambda \geq \max\{1, c_6^2 L\}$, we have

$$\sum_{i=1}^{t} \|\nabla f_{\boldsymbol{W}^{(i-1)}}(\boldsymbol{x}_{i,a_i}) \cdot m^{-1/2}\|_{\boldsymbol{\Lambda}_{i-1}^{-1}}^2 \leq 2 \log \frac{\det(\boldsymbol{\Lambda}_t)}{\det(\lambda \boldsymbol{I})}.$$

For the second part of Lemma B.5, for any $t \in [n]$, we define

$$\bar{\boldsymbol{M}}_t = m^{-1/2} \cdot [\text{vec}(\nabla f_{\boldsymbol{W}^{(0)}}(\boldsymbol{x}_{1,a_1})), \dots, \text{vec}(\nabla f_{\boldsymbol{W}^{(0)}}(\boldsymbol{x}_{t,a_t}))] \in \mathbb{R}^{p \times t},$$
$$\boldsymbol{M}_t = m^{-1/2} \cdot [\text{vec}(\nabla f_{\boldsymbol{W}^{(t-1)}}(\boldsymbol{x}_{1,a_1})), \dots, \text{vec}(\nabla f_{\boldsymbol{W}^{(t-1)}}(\boldsymbol{x}_{t,a_t}))] \in \mathbb{R}^{p \times t}.$$

We have $\bar{\boldsymbol{\Lambda}}_t = \lambda \boldsymbol{I} + \bar{\boldsymbol{M}}_t \bar{\boldsymbol{M}}_t^T$ and $\boldsymbol{\Lambda}_t = \lambda \boldsymbol{I} + \boldsymbol{M}_t \boldsymbol{M}_t^T$, and

$$\left| \log \frac{\det(\boldsymbol{\Lambda}_t)}{\det(\lambda \boldsymbol{I})} - \log \frac{\det(\bar{\boldsymbol{\Lambda}}_t)}{\det(\lambda \boldsymbol{I})} \right| = |\log \det(\boldsymbol{I} + \boldsymbol{M}_t \boldsymbol{M}_t^T/\lambda) - \log \det(\boldsymbol{I} + \bar{\boldsymbol{M}}_t \bar{\boldsymbol{M}}_t^T/\lambda)|$$

$$= |\log \det(\boldsymbol{I} + \boldsymbol{M}_t^T \boldsymbol{M}_t/\lambda) - \log \det(\boldsymbol{I} + \bar{\boldsymbol{M}}_t^T \bar{\boldsymbol{M}}_t/\lambda)|$$

$$\leq \max\{\|\langle(\boldsymbol{I} + \boldsymbol{M}_t^T \boldsymbol{M}_t/\lambda)^{-1}, \boldsymbol{M}_t^T \boldsymbol{M}_t - \bar{\boldsymbol{M}}_t^T \bar{\boldsymbol{M}}_t\rangle\|, \|\langle(\boldsymbol{I} + \bar{\boldsymbol{M}}_t^T \bar{\boldsymbol{M}}_t/\lambda)^{-1}, \boldsymbol{M}_t^T \boldsymbol{M}_t - \bar{\boldsymbol{M}}_t^T \bar{\boldsymbol{M}}_t\rangle\|\}$$

$$= \|\langle(\boldsymbol{I} + \boldsymbol{M}_t^T \boldsymbol{M}_t/\lambda)^{-1}, \boldsymbol{M}_t^T \boldsymbol{M}_t - \bar{\boldsymbol{M}}_t^T \bar{\boldsymbol{M}}_t\rangle\|$$

$$\leq \|(\boldsymbol{I} + \boldsymbol{M}_t^T \boldsymbol{M}_t/\lambda)^{-1}\|_F \cdot \|\boldsymbol{M}_t^T \boldsymbol{M}_t - \bar{\boldsymbol{M}}_t^T \bar{\boldsymbol{M}}_t\|_F$$

$$\leq \sqrt{t}\|(\boldsymbol{I} + \boldsymbol{M}_t^T \boldsymbol{M}_t/\lambda)^{-1}\|_2 \cdot \|\boldsymbol{M}_t^T \boldsymbol{M}_t - \bar{\boldsymbol{M}}_t^T \bar{\boldsymbol{M}}_t\|_F$$

$$\leq \sqrt{t}\|\boldsymbol{M}_t^T \boldsymbol{M}_t - \bar{\boldsymbol{M}}_t^T \bar{\boldsymbol{M}}_t\|_F$$

$$\leq \sqrt{t} t \max_{1 \leq i,j \leq t} m^{-1} \cdot \left| \langle\nabla f_{\boldsymbol{W}^{(t-1)}}(\boldsymbol{x}_{i,a_i}), \nabla f_{\boldsymbol{W}^{(t-1)}}(\boldsymbol{x}_{j,a_j})\rangle - \langle\nabla f_{\boldsymbol{W}^{(0)}}(\boldsymbol{x}_{i,a_i}), \nabla f_{\boldsymbol{W}^{(0)}}(\boldsymbol{x}_{j,a_j})\rangle \right|$$

$$\leq t^{3/2} m^{-1} \max_{1 \leq i,j \leq t} \left( \left| \langle\nabla f_{\boldsymbol{W}^{(t-1)}}(\boldsymbol{x}_{i,a_i}) - \nabla f_{\boldsymbol{W}^{(0)}}(\boldsymbol{x}_{i,a_i}), \nabla f_{\boldsymbol{W}^{(t-1)}}(\boldsymbol{x}_{j,a_j})\rangle \right| \right.$$

$$+ \left. \left| \langle\nabla f_{\boldsymbol{W}^{(t-1)}}(\boldsymbol{x}_{j,a_j}) - \nabla f_{\boldsymbol{W}^{(0)}}(\boldsymbol{x}_{j,a_j}), \nabla f_{\boldsymbol{W}^{(t-1)}}(\boldsymbol{x}_{i,a_i})\rangle \right| \right)$$

$$\leq 2t^{3/2} m^{-1} \max_i \|\nabla f_{\boldsymbol{W}^{(t-1)}}(\boldsymbol{x}_{i,a_i}) - \nabla f_{\boldsymbol{W}^{(0)}}(\boldsymbol{x}_{i,a_i})\|_F \cdot \max_i \|\nabla f_{\boldsymbol{W}^{(t-1)}}(\boldsymbol{x}_{i,a_i})\|_F$$

$$\leq 2C_2 C_3^2 t^{3/2} m^{-1/6} (\log m)^{1/2} L^{23/6} \lambda^{-1/6}$$

where the second equality is by $\det(\boldsymbol{I} + \boldsymbol{A}\boldsymbol{A}^T) = \det(\boldsymbol{I} + \boldsymbol{A}^T\boldsymbol{A})$, the first inequality is by that $\log \det$ is concave, the third equality is assumed without loss of generality, the second inequality is by that $\langle\boldsymbol{A}, \boldsymbol{B}\rangle \leq \|\boldsymbol{A}\|_F \|\boldsymbol{B}\|_F$, the third inequality is by that $\|\boldsymbol{A}\|_F \leq \sqrt{t}\|\boldsymbol{A}\|_2$ for $\boldsymbol{A} \in \mathbb{R}^{t \times t}$, the fourth inequality is by that $\boldsymbol{I} + \boldsymbol{M}_t^T \boldsymbol{M}_t/\lambda \succeq \boldsymbol{I}$, the fifth inequality is by that $\|\boldsymbol{A}\|_F \leq t\|\boldsymbol{A}\|_\infty$ for $\boldsymbol{A} \in \mathbb{R}^{t \times t}$, the sixth inequality is by the triangle inequality, and the last inequality is by Lemma B.3, and Lemma C.2.

The third part of Lemma B.5 directly follows the argument in (**?**, (B.18)) and uses Lemma C.1 for $\epsilon = (nK)^{-1}$.

Finally, it is easy to verify that the condition of $m$ in Lemma B.5 satisfies the condition of $m$ for $\mathcal{E}(\delta/(n+2))$, and union bound we have $\mathbb{P}(\mathcal{E}(\delta/(n+2))) \geq 1 - \delta$. □

## D  DETAILS OF BASELINE ALGORITHMS IN SECTION 6

In this section, we present the details of each representative baseline methods and of the B-mode version of NeuraLCB in Section 6. We summarize the baseline methods in Table 2.

Table 2: A summary of the baseline methods.

| Baseline | Algorithm | Type | Function approximation |
|---|---|---|---|
| LinLCB | Algorithm 2 | Pessimism | Linear |
| KernLCB | Algorithm 3 | Pessimism | RKSH |
| NeuralLinLCB | Algorithm 4 | Pessimism | Linear w/ fixed neural features |
| NeuralLinGreedy | Algorithm 5 | Greedy | Linear w/ fixed neural features |
| NeuralGreedy | Algorithm 6 | Greedy | Neural networks |

---

**Algorithm 2 LinLCB**

---

**Input:** Offline data $\mathcal{D}_n = \{(\boldsymbol{x}_t, a_t, r_t)\}_{t=1}^n$, reg. parameter $\lambda > 0$, confidence parameters $\beta > 0$.
1: $\boldsymbol{\Lambda}_n \leftarrow \lambda \boldsymbol{I} + \sum_{t=1}^n \boldsymbol{x}_{t,a_t} \boldsymbol{x}_{t,a_t}^T$
2: $\hat{\boldsymbol{\theta}}_n \leftarrow \boldsymbol{\Lambda}_n^{-1} \sum_{t=1}^n \boldsymbol{x}_{t,a_t} r_t$
3: $L(\boldsymbol{u}) \leftarrow \langle \hat{\boldsymbol{\theta}}_n, \boldsymbol{u} \rangle - \beta \|\boldsymbol{u}\|_{\boldsymbol{\Lambda}_n^{-1}}, \forall \boldsymbol{u} \in \mathbb{R}^d$
**Output:** $\hat{\pi}(\boldsymbol{x}) \leftarrow \arg\max_{a \in [K]} L(\boldsymbol{x}_a)$

---

---

**Algorithm 3 KernLCB**

---

**Input:** Offline data $\mathcal{D}_n = \{(\boldsymbol{x}_t, a_t, r_t)\}_{t=1}^n$, reg. parameter $\lambda > 0$, confidence parameters $\beta > 0$, kernel $k : \mathbb{R}^d \times \mathbb{R}^d \to \mathbb{R}$.
1: $\boldsymbol{k}_n(\boldsymbol{u}) \leftarrow [k(\boldsymbol{u}, \boldsymbol{x}_{1,a_1}), \ldots, k(\boldsymbol{u}, \boldsymbol{x}_{n,a_n})]^T, \forall \boldsymbol{u} \in \mathbb{R}^d$
2: $\boldsymbol{K}_n \leftarrow [k(\boldsymbol{x}_{i,a_i}, \boldsymbol{x}_{j,a_j})]_{1 \leq i,j \leq n}$
3: $\boldsymbol{y}_n \leftarrow [r_1, \ldots, r_n]^T$
4: $L(\boldsymbol{u}) \leftarrow \boldsymbol{k}_n(\boldsymbol{u})^T (\boldsymbol{K}_n + \lambda I)^{-1} \boldsymbol{y}_n - \beta \sqrt{k(\boldsymbol{u}, \boldsymbol{u}) - \boldsymbol{k}_n^T(\boldsymbol{u})(\boldsymbol{K}_n + \lambda I)^{-1} \boldsymbol{k}_n}, \forall \boldsymbol{u} \in \mathbb{R}^d$
**Output:** $\hat{\pi}(\boldsymbol{x}) \leftarrow \arg\max_{a \in [K]} L(\boldsymbol{x}_a)$

---

---

**Algorithm 4 NeuralLinLCB**

---

**Input:** Offline data $\mathcal{D}_n = \{(\boldsymbol{x}_t, a_t, r_t)\}_{t=1}^n$, regularization parameter $\lambda > 0$, confidence parameters $\beta > 0$.
1: Initialize the same neural network with the same initialization scheme as in NeuraLCB to obtain the neural network function $f_{\boldsymbol{W}^{(0)}}$ at initialization
2: $\phi(\boldsymbol{u}) \leftarrow \text{vec}(\nabla f_{\boldsymbol{W}^{(0)}}(\boldsymbol{u})) \in \mathbb{R}^p, \forall \boldsymbol{u} \in \mathbb{R}^d$
3: $\boldsymbol{\Lambda}_n \leftarrow \lambda \boldsymbol{I} + \sum_{t=1}^n \phi(\boldsymbol{x}_{t,a_t}) \phi(\boldsymbol{x}_{t,a_t})^T$
4: $\hat{\boldsymbol{\theta}}_n \leftarrow \boldsymbol{\Lambda}_n^{-1} \sum_{t=1}^n \phi(\boldsymbol{x}_{t,a_t}) r_t$
5: $L(\boldsymbol{u}) \leftarrow \langle \hat{\boldsymbol{\theta}}_n, \phi(\boldsymbol{u}) \rangle - \beta \|\phi(\boldsymbol{u})\|_{\boldsymbol{\Lambda}_n^{-1}}, \forall \boldsymbol{u} \in \mathbb{R}^d$
**Output:** $\hat{\pi}(\boldsymbol{x}) \leftarrow \arg\max_{a \in [K]} L(\boldsymbol{x}_a)$

---

---

**Algorithm 5 NeuralLinGreedy**

---

**Input:** Offline data $\mathcal{D}_n = \{(\boldsymbol{x}_t, a_t, r_t)\}_{t=1}^n$, regularization parameter $\lambda > 0$.

1: Initialize the same neural network with the same initialization scheme as in NeuraLCB to obtain the neural network function $f_{\boldsymbol{W}^{(0)}}$ at initialization
2: $\phi(\boldsymbol{u}) \leftarrow \text{vec}(\nabla f_{\boldsymbol{W}^{(0)}}(\boldsymbol{u})) \in \mathbb{R}^p, \forall \boldsymbol{u} \in \mathbb{R}^d$
3: $\hat{\boldsymbol{\theta}}_n \leftarrow \boldsymbol{\Lambda}_n^{-1} \sum_{t=1}^n \phi(\boldsymbol{x}_{t,a_t}) r_t$
4: $L(\boldsymbol{u}) \leftarrow \langle \hat{\boldsymbol{\theta}}_n, \phi(\boldsymbol{u}) \rangle, \forall \boldsymbol{u} \in \mathbb{R}^d$

**Output:** $\hat{\pi}(\boldsymbol{x}) \leftarrow \arg\max_{a \in [K]} L(\boldsymbol{x}_a)$

---

**Algorithm 6 NeuralGreedy**

---

**Input:** Offline data $\mathcal{D}_n = \{(\boldsymbol{x}_t, a_t, r_t)\}_{t=1}^n$, step sizes $\{\eta_t\}_{t=1}^n$, regularization parameter $\lambda > 0$.

1: Initialize $\boldsymbol{W}^{(0)}$ as follows: set $\boldsymbol{W}_l^{(0)} = [\bar{\boldsymbol{W}}_l, \boldsymbol{0}; \boldsymbol{0}, \bar{\boldsymbol{W}}_l], \forall l \in [L-1]$ where each entry of $\bar{\boldsymbol{W}}_l$ is generated independently from $\mathcal{N}(0, 4/m)$, and set $\boldsymbol{W}_L^{(0)} = [\boldsymbol{w}^T, -\boldsymbol{w}^T]$ where each entry of $\boldsymbol{w}$ is generated independently from $\mathcal{N}(0, 2/m)$.
2: **for** $t = 1, \ldots, n$ **do**
3:      Retrieve $(\boldsymbol{x}_t, a_t, r_t)$ from $\mathcal{D}_n$.
4:      $\hat{\pi}_t(\boldsymbol{x}) \leftarrow \arg\max_{a \in [K]} f_{\boldsymbol{W}^{(t-1)}}(\boldsymbol{x}_a), \forall \boldsymbol{x}$.
5:      $\boldsymbol{W}^{(t)} \leftarrow \boldsymbol{W}^{(t-1)} - \eta_t \nabla \mathcal{L}_t(\boldsymbol{W}^{(t-1)})$ where $\mathcal{L}_t(\boldsymbol{W}) = \frac{1}{2}(f_{\boldsymbol{W}}(\boldsymbol{x}_{t,a_t}) - r_t)^2 + \frac{m\lambda}{2}\|\boldsymbol{W} - \boldsymbol{W}^{(0)}\|_F^2$.
6: **end for**

**Output:** Randomly sample $\hat{\pi}$ uniformly from $\{\hat{\pi}_1, \ldots, \hat{\pi}_n\}$.

---

**Algorithm 7 NeuraLCB (B-mode)**

---

**Input:** Offline data $\mathcal{D}_n = \{(\boldsymbol{x}_t, a_t, r_t)\}_{t=1}^n$, step sizes $\{\eta_t\}_{t=1}^n$, regularization parameter $\lambda > 0$, confidence parameters $\{\beta_t\}_{t=1}^n$, batch size $B > 0$, epoch number $J > 0$.

1: Initialize $\boldsymbol{W}^{(0)}$ as follows: set $\boldsymbol{W}_l^{(0)} = [\bar{\boldsymbol{W}}_l, \boldsymbol{0}; \boldsymbol{0}, \bar{\boldsymbol{W}}_l], \forall l \in [L-1]$ where each entry of $\bar{\boldsymbol{W}}_l$ is generated independently from $\mathcal{N}(0, 4/m)$, and set $\boldsymbol{W}_L^{(0)} = [\boldsymbol{w}^T, -\boldsymbol{w}^T]$ where each entry of $\boldsymbol{w}$ is generated independently from $\mathcal{N}(0, 2/m)$.
2: $\boldsymbol{\Lambda}_0 \leftarrow \lambda \boldsymbol{I}$.
3: **for** $t = 1, \ldots, n$ **do**
4:      Retrieve $(\boldsymbol{x}_t, a_t, r_t)$ from $\mathcal{D}_n$.
5:      $L_t(\boldsymbol{u}) \leftarrow f_{\boldsymbol{W}^{(t-1)}}(\boldsymbol{u}) - \beta_{t-1} \|\nabla f_{\boldsymbol{W}^{(t-1)}}(\boldsymbol{u}) \cdot m^{-1/2}\|_{\boldsymbol{\Lambda}_{t-1}^{-1}}, \forall \boldsymbol{u} \in \mathbb{R}^d$
6:      $\hat{\pi}_t(\boldsymbol{x}) \leftarrow \arg\max_{a \in [K]} L_t(\boldsymbol{x}_a)$, for all $\boldsymbol{x} = \{\boldsymbol{x}_a \in \mathbb{R}^d : a \in [K]\}$.
7:      $\boldsymbol{\Lambda}_t \leftarrow \boldsymbol{\Lambda}_{t-1} + \text{vec}(\nabla f_{\boldsymbol{W}^{(t-1)}}(\boldsymbol{x}_{t,a_t})) \cdot \text{vec}(\nabla f_{\boldsymbol{W}^{(t-1)}}(\boldsymbol{x}_{t,a_t}))^T / m$.
8:      $\tilde{\boldsymbol{W}}^{(0)} \leftarrow \boldsymbol{W}^{(t-1)}$
9:      **for** $j = 1, \ldots, J$ **do**
10:          Sample a batch of data $B_t = \{\boldsymbol{x}_{t_q,a_{t_q}}, r_{t_q}\}_{q=1}^B$ from $\mathcal{D}_t$
11:          $\mathcal{L}_t^{(j)}(\boldsymbol{W}) \leftarrow \sum_{q=1}^B \frac{1}{2B}(f_{\boldsymbol{W}}(\boldsymbol{x}_{t_q,a_{t_q}}) - r_{t_q})^2 + \frac{m\lambda}{2}\|\boldsymbol{W} - \boldsymbol{W}^{(0)}\|_F^2$
12:          $\tilde{\boldsymbol{W}}^{(j)} \leftarrow \tilde{\boldsymbol{W}}^{(j-1)} - \eta_t \nabla \mathcal{L}_t^{(j)}(\tilde{\boldsymbol{W}}^{(j-1)})$
13:      **end for**
14:      $\boldsymbol{W}^{(t)} \leftarrow \tilde{\boldsymbol{W}}^{(J)}$
15: **end for**

**Output:** Randomly sample $\hat{\pi}$ uniformly from $\{\hat{\pi}_1, \ldots, \hat{\pi}_n\}$.

---

# E  DATASETS

We present a detailed description about the UCI datasets used in our experiment.

- **Mushroom data**: Each sample represents a set of attributes of a mushroom. There are two actions to take on each mushroom sample: eat or no eat. Eating an editable mushroom

Table 3: The real-world dataset statistics

| Dataset | Mushroom | Statlog | Adult | MNIST |
|---|---|---|---|---|
| Context dimension | 22 | 9 | 94 | 784 |
| Number of classes | 2 | 7 | 14 | 10 |
| Number of instances | 8,124 | 43,500 | 45,222 | 70,000 |

     generates a reward of $+5$ while eating a poisonous mushroom yields a reward of $+5$ with probability $0.5$ and a reward of $-35$ otherwise. No eating gives a reward of $0$.

- **Statlog data**: The shuttle dataset contains the data about a space shuttle flight where the goal is to predict the state of the radiator subsystem of the shuttle. There are total $K = 7$ states to predict where approximately $80\%$ of the data belongs to one state. A learner receives a reward of $1$ if it selects the correct state and $0$ otherwise.

- **Adult data**: The Adult dataset contains personal information from the US Census Bureau database. Following (Riquelme et al., 2018), we use the $K = 14$ different occupations as actions and $d = 94$ covariates as contexts. As in the Statlog data, a learner obtains a reward of $1$ for making the right prediction, and $0$ otherwise.

- MNIST data: The MNIST data contains images of various handwritten digits from $0$ to $9$. We use $K = 10$ different digit classes as actions and $d = 784$ covariates as contexts. As in the Statlog and Adult data, a learner obtains a reward of $1$ for making the right prediction, and $0$ otherwise.

We summarizes the statistics of the above datasets in Table 3.

## F   ADDITIONAL EXPERIMENTS

In this section, we complement the experimental results in the main paper with additional experiments regarding the learning ability of our algorithm on dependent data and the different behaviours of S-mode and B-mode training.

### F.1   DEPENDENT DATA

As the sub-optimality guarantee in Theorem 4.1 does not require the offline policy to be stationary, we evaluate the empirical performance of our algorithm and the baselines on a new setup of offline data collection that represents dependent actions. In particular, instead of using a stationary policy to collect offline actions as in Section 6, here we used an adaptive policy $\mu$ defined as

$$\mu(a|\mathcal{D}_{t-1}, \boldsymbol{x}_t) = (1 - \epsilon) * \pi^*(a|\boldsymbol{x}_t) + \epsilon * \pi_{\text{LinUCB}}(a|\mathcal{D}_{t-1}, \boldsymbol{x}_t),$$

where $\pi^*$ is the optimal policy and $\pi_{\text{LinUCB}}$ is the linear UCB learner (Abbasi-Yadkori et al., 2011). This weighted policy makes the collected offline data $a_t$ dependent on $\mathcal{D}_{t-1}$ while making sure that the offline data has a sufficient coverage over the data of the optimal policy, as LinUCB does not perform well in several non-linear data. We used $\epsilon = 0.9$ in this experiment.

The results on Statlog and MNIST are shown in Figure 3. We make two important observations. First, on this dependent data, the baseline methods with linear models (LinLCB, NeuralLinLCB and NeuralLinGreedy) [3] perform almost as bad as when they learn on the independent data in Figure 2, suggesting that linear models are highly insufficient to learn complex rewards in real-world data, regardless of how offline data were collected. Secondly, the main competitor of our method, NeuralGreedy suffers an apparent performance degradation (especially in a larger dataset like MNIST) while NeuraLCB maintains a superior performance, suggesting the effectiveness of pessimism in our method on dealing with offline data and the robustness of our method toward dependent data.

---

[3] KernLCB also does not change its performance much, but its computational complexity is still an major issue.

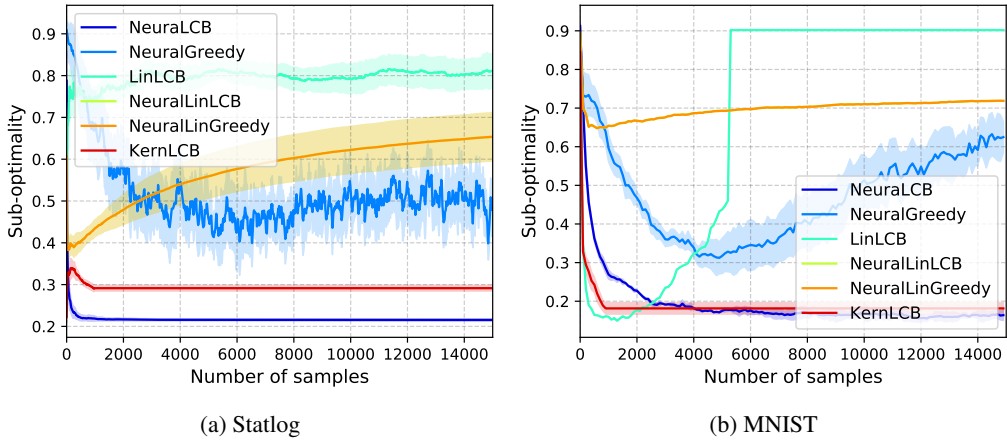

(a) Statlog                    (b) MNIST

Figure 3: The sub-optimality of NeuraLCB versus the baseline algorithms on real-world datasets with correlated structures.

### F.2    S-MODE VERSUS B-MODE TRAINING

As in Section 6 we implemented two different training modes: S-mode (Algorithm 1) and B-mode (Algorithm 7). We compare the empirical performances of S-mode and B-mode on various datasets. As this variant is only applicable to NeuralGreedy and NeuraLCB, we only depict the performances of these algorithms. The results on Cosine (synthetic dataset), Statlog, Adult and MNIST are shown in Figure 4.

We make the following observations, which are consistent with the conclusion in the main paper while giving more insights. While the B-mode outperforms the S-mode on Cosine, the S-mode significantly outperforms the B-mode in all the tested real-world datasets. Moreover, the B-mode of NeuraLCB outperforms or at least is compatible to the S-mode of NeuralGreedy in the real-world datasets. First, these observations suggest the superiority of our method on the real-world datasets. Second, to explain the performance difference of S-mode and B-mode on synthetic and real-world datasets in our experiment, we speculate that the online-like nature of our algorithm tends to reduce the need of B-mode in practical datasets because B-mode in the streaming data might cause over-fitting (as there could be some data points in the past streaming that has been fitted for multiple times). In synthetic and simple data such as Cosine, over-fitting tends to associate with strong prediction of the underlying reward function as the simple synthetic reward function such as Cosine is sufficiently smooth, unlike practical reward functions in the real-world datasets.

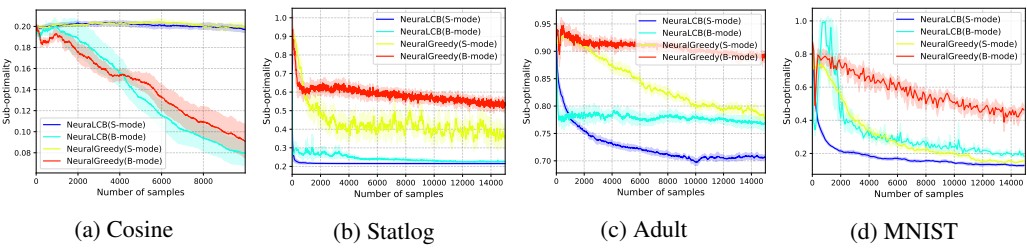

(a) Cosine          (b) Statlog          (c) Adult          (d) MNIST

Figure 4: Comparison of S-mode and B-mode training.

