# OpenReview forum: "Offline Neural Contextual Bandits: Pessimism, Optimization and Generalization"
_ICLR.cc/2022/Conference — ICLR 2022 Poster_

### Official Review · Reviewer_HWn2 · 2021-10-24

**Correctness:** 3
**Technical Novelty And Significance:** 2
**Empirical Novelty And Significance:** 2
**Recommendation:** 6
**Confidence:** 3

**Main Review:**

Pros:

1. The pessimism principle (Kidambi et al., 2020; Buckman et al., 2020; Jin et al., 2020) is a recent idea introduced in off-policy learning to avoid the strong assumption of sufficient coverage of the data. The key idea is to consider a regularization version, i.e. lower confidence bound (LCB). After its introduction, it has been applied in various off-policy RL and off-policy bandit settings. This paper further advances this area by introducing this idea into the neural contextual bandit setting.


2. This paper is very well written and is easy to follow.


Cons:

1. Related to my previous comments on the pessimism principle, the main contribution of this paper is to incorporate the pessimism principle neural contextual bandit setting. It combines the strength of both Rashidinejad et al. (2021) and Zhou et al. (2020), where the former studies the lower confidence bound (pessimism principle) for decision-making in tabular MDP and contextual bandit, and the latter studies neural contextual bandit with upper confidence bound in typical online setting. Therefore, the technical advantages over these existing work are expected and routine.

2. In Algorithm 1, why to randomly sample policy uniformly from $\hat{\pi}_1, \ldots, \hat{\pi}_n$?

3. In Theorem 4.1, the error bound $O(\kappa \tilde{d}^{1/2} n^{-1/2})$, $\kappa$ measures the distributional shift and $\tilde{d}$ is an effective dimension of the neural network.

(1) The Assumption 4.2 requires $\kappa$ to be a uniform upper bound over all sample size $n$ and all samples $x_t$. $\kappa$ would be a function of the dimension and the sample size. In this case, the error bound might diverge. It would be more convincing if the authors could provide some justifications or examples when $\kappa$ would be small.

(2) Similarly, the effective dimension of the neural network $\tilde{d}$ might be very large. This effective dimension of the neural network was used for online neural contextual bandit (Zhou et al., 2019) and neural MDP (Yang et al., 2020). The authors commented that $\tilde{d}$ in these references were typically small, in the order of $log(n)$. However, it is unknown whether  $\tilde{d}$ in the proposed off-policy neural contextual bandit is still small. In order to ensure this, one would inevitably assume conditions on the data generation process to control the decaying speed of eigenvalues of the gram matrix $H$. It would be more convincing if the authors could provide detailed quantification of data generation process.


**Summary Of The Paper:**

~~~~~~~~~~~~~~
Thank the authors for the clarification. Some of my former comments were nicely addressed and hence I am willing to increase my rating to 6.

~~~~~~~~~~~~~~
This paper considers the offline setting of the contextual bandit with neural network function approximation. The key idea of the proposed NeuraLCB is to use neural network to learn the reward function and use a pessimism principle via a lower confidence bound (LCB) for decision making. In theory, the proposed approach is shown to learn the optimal policy with an error bound $O(\kappa \tilde{d}^{1/2} n^{-1/2})$ where $\kappa$ measures the distributional shift and $\tilde{d}$ is an effective dimension of the neural network. The empirical effectiveness of the proposed method is shown in a range of synthetic and real-world off-policy learning problems.



**Summary Of The Review:**

The technical contribution of this paper is OK but not strong. The assumptions and theory of the paper need substantial clarifications.

---

> ### Author Response · Authors · 2021-11-16
> **Response**
>
> Thank you for your comments. Below we address all your concerns. Please kindly let us know if you have any further questions.
>
> ### **Q1: “the technical advantages over these existing work are expected and routine [because it combined pessimism and neural contextual bandits]”**
>
> While we did use pessimism and neural networks in our offline learning setting, we would like to highlight the technical advantages of our work over these existing works. That is, we established a practical algorithm in neural network function approximation setting with a strong theoretical guarantee that requires milder data coverage assumption (Assumption 4.2) than any prior offline algorithm (e.g., Rashidinejad et al. (2021), see in Table 1), reduces the optimization complexity from $\mathcal{O}(n^2)$ to $\mathcal{O}(n)$, and improves the dependence on $\tilde{d}$. These achievements are particularly significant for offline policy learning as the prior works establish theoretical guarantees in much more restricted settings (e.g., tabular representation, linear models, uniform data coverage assumption, or intractable algorithms) while our results are stronger and applicable to broader real-world setting requiring milder assumptions. Thus, we believe that our paper significantly advances the capability of offline policy learning compared to existing state-of-the-art works.
>
> ### **Q2: “why to randomly sample policy uniformly from $\hat{\pi}_1,…,\hat{\pi}_n$?”**
>
> Thanks to the uniform sampling, we have $\mathbb{E}[SubOpt(\hat{\pi})] = \frac{1}{n} \sum_{t=1}^n SubOpt(\hat{\pi}_t)$ which was used to derive a generalization bound to unseen contexts in the proof of Lemma A.3. Due to this argument (and the online-like nature of our algorithm), we can avoid any uniform convergence bound which is commonly used in offline policy learning. For complex models such as neural networks, a uniform convergence bound is unnecessarily large or even vacuous.
>
> ### **Q3: “$\kappa$ would be a function of the dimension and the sample size. In this case, the error bound might diverge”**
>
> In our paper, we assumed that $\kappa$ is a finite constant (independent of dimension and sample size). A sufficient condition (though stronger than necessary) for this assumption is when the behavior policy has sufficient coverage over the optimal policy in all contexts. As a concrete example, consider the offline policy to be an $\epsilon$-greedy w.r.t. the optimal policy (i.e., it takes an optimal action with a probability of $1-\epsilon$, and a uniformly random action with a probability of $\epsilon$). In this case, we can choose $\kappa = \frac{1}{1 - \epsilon + \epsilon/K}$, which is independent of both the dimension and the sample size.
>
> ### **Q4: “it is unknown whether $\tilde{d}$ in the proposed off-policy neural contextual bandit is still small”**
> We remark that $\tilde{d}$ is defined through the NTK matrix $H$ which is in turn defined on the observed full contexts of **all** actions, not just the offline actions (see Definition 4.1). Thus, along with the first part of Assumption 4.2 (that the full contexts {$x_t$} are generated independent of any policy, which is standard and minimal in online setting, e.g., {${x}_t$} $\overset{i.i.d.}{\sim} \rho$ (Lattimore & Szepesvari, 2020; Rashidinejad et al., 2021; Papini et al., 2021)), $\tilde{d}$ in our setting does not depend on the offline policy, and it is identical to the effective dimension used in an online setting (e.g., (Zhou et al., 2019)), which only depends on $n$ logarithmically for most interesting cases (Yang et al., 2020).

---

> > ### Author Response · Authors · 2021-12-01
> > **Thank you**
> >
> > We thank you for acknowledging our response and revision and updating your score.

---

### Official Review · Reviewer_RvBs · 2021-11-02

**Correctness:** 3
**Technical Novelty And Significance:** 2
**Empirical Novelty And Significance:** 2
**Recommendation:** 6
**Confidence:** 3

**Main Review:**

Pros:

a) The paper provides concrete theoretical analysis to support the proposed algorithm. I have not gone through all the derivations, but the overall result looks good.

b) Comprehensive comparisons with other related works are properly presented.

c) The paper is generally well written. The required assumptions are discussed clearly.

Cons:

a) Apart from theoretical analysis, it would be better if the paper could throw some light on algorithm design consideration beyond LCB-like algorithm.

b) It is not very clear how the improvement of \sqrt{d} and O(n) is achieved. It would be better to give more concrete discussions.


**Summary Of The Paper:**

The paper studies the problem of offline contextual bandits, where policy learning can only leverage a fixed dataset collected a priori by behavior policies. Using a pessimism principle, the authors propose a new algorithm called NeuralLCB with overparameterized neural networks and provide theoretical regret guarantees based on the analysis framework of the neural tangent kernel. Experiments on both synthetic and real-world data are conducted, which confirms the theoretical results.

**Summary Of The Review:**

Overall the paper studies a new problem and presents a good analysis. Since the technique is quite similar to existing works, it would be much better to present necessary discussions to claim their contributions.

---

> ### Author Response · Authors · 2021-11-16
> **Response**
>
> Thank you for your comments. Below we address all your concerns. Please kindly let us know if you have any further questions.
>
> ### **Q1: “some light on algorithm design consideration beyond LCB-like algorithm”**
>
> Beyond LCB, the two key design considerations in our algorithm are (i) the use of “online-like” learning for offline learning (one may think of it as a data splitting technique to ensure generalization), and (ii) the adaptive update of one single neural network at each iteration (instead of training new network from scratch at each iteration as in Zhou et al. (2020)). While (i) can guarantee generalization beyond observed contexts, (ii) reduces the computational complexity from $\mathcal{O}(n^2)$ to $\mathcal{O}(n)$.
>
> ### **Q2: “It is not very clear how the improvement of \sqrt{d} and O(n) is achieved”**
>
> * **Regarding the improvement on $\sqrt{\tilde{d}}$**: On a technical side, the key distinction with Zhou et al. (2020) is that we regress toward the optimal parameters $W^*$ of the neural network (those that interpolate the training data) rather than toward the empirical risk minimizer as in Zhou et al. (2020). This idea leads to two important design benefits in our algorithms: (i) we can learn the data in an streaming manner, one data point at a time , and (ii) we can remove the dependence of the confidence parameter $\beta_t$ on $\sqrt{\tilde{d}}$. Here, (ii) is the key to save us a cost of $\sqrt{\tilde{d}}$ in our bound.
>
> * **Regarding the computational improvement on $\mathcal{O}(n)$**: Our algorithm adaptively updates one single neural network on one data point at a time in each iteration $t$, while Zhou et al. (2020) trains a new neural network from scratch in $\mathcal{D}_{t-1}$ for $\mathcal{O}(t)$ epochs at each iteration $t$. Thus, our algorithm computation only scales with $\mathcal{O}(n)$ while Zhou et al. (2020) scales with $\mathcal{O}(n^2)$.

---

### Official Review · Reviewer_5NdR · 2021-11-03

**Correctness:** 3
**Technical Novelty And Significance:** 3
**Empirical Novelty And Significance:** 2
**Recommendation:** 6
**Confidence:** 4

**Main Review:**

Strength

- Comparing with existing works, NeuraLCB requires weaker assumptions on data coverage (empirical single-policy concentration condition) and data generation (data can be dependent on history / past data) and I think this is a major contribution of the paper. The weaker assumption makes the algorithm applicable to more practical settings.

- Theoretical analysis on sub-optimality learned by NeuraLCB is provided. The algorithm essentially works in an online fashion that trained on one data point at each iteration. Thus several intermediate results can be directly applied from or very similar to online regret minimization in neural bandits (e.g., NeuralUCB, Zhou et al., 2019). If my understanding is correct, due to the different goal between online and offline policy learning (instead of improved technical lemmas over Zhou et al., 2019), the design of NeuraLCB is very different from NeuralUCB, e.g., single data point SGD verse full gradient descent at every iteration.

I checked the main theorem and several lemmas; they look sound and I only found a few mistakes (see below).

Weakness

- Mistakes in proof of Lemma B.5:
  - $\log \det$ function is concave instead of convex.
  - When bounding the difference of the $\log \det$ functions, the first inequality does not hold. Because log det function is concave, by taylor expansion we have that $\log \det(X) - \log \det(Y) \leq <Y^{-1}, X-Y>_F$. But the inequality does not hold when taking the absolute value of both sides, which is the case in current proof. This problem should be fixable and Lemma B.5 should still hold.

- In experiments the authors mentioned a B-mode variant of NeuraLCB (and NeuralGreedy) that update with a small batch of data points at each iteration. The result is reported after grid search over B-mode and S-mode (one step SGD on single data point). I would suggest the authors to report the performance of S-mode and B-mode separately. In experiment on the real-world datasets, the authors observed that S-mode performed better than B-mode (again, both results should be plotted). That's an interesting observation. It would be helpful to provide some explanations over this observation.

Other comments

- Besides the theoretical convenience of learning in an online manner, are there any other reasons to prevent full gradient descent training? It sounds very natural for offline learning.

- The authors mentioned after Assumption 4.2 that NeuraLCB works with depend data: 'the offline data was collected by an active learner such as a Q-learning agent'. Currently the synthetic experiment is on independent data. It would be interesting to see if NeuraLCB could work well with depend data, e.g., use some bandit algorithms to collect the dataset.




**Summary Of The Paper:**

This paper is the first study considers offline policy learning for contextual bandits with neural networks. The authors proposed NeuraLCB algorithm that used neural network to model the rewards and followed pessimism principle with lower confidence bound in policy learning. It is a very intuitive combination. The algorithm works with mild assumptions with theoretical guarantee on its suboptimality. Experiments also showed that NeuraLCB outperforms other baselines.

**Summary Of The Review:**

Overall I think this is a good paper with a straightforward idea of combining pessimism and neural bandits for offline policy learning. I am open to revise my rating if the authors could address the concerns.

---

> ### Author Response · Authors · 2021-11-16
> **Response**
>
> Thank you for your suggestions. Below we address all your concerns. Please kindly let us know if you have any further questions.
>
> ### **Q1: “due to the different goal between online and offline policy learning (instead of improved technical lemmas over Zhou et al., 2019), the design of NeuraLCB is very different from NeuralUCB, e.g., single data point SGD verse full gradient descent at every iteration.”**
>
> We remark that the optimization design in our algorithm (single data point SGD) is of independent interest that does not only apply to the offline setting but also to the original online setting in (Zhou et al., 2019) to improve their regret and optimization complexity. We added this remark to our revised paper.
>
> ### **Q2: The fixable mistake in the proof of Lemma B.5**
> Thank you for pointing out this detailed mistake. As we have $|\log det (X) - \log det(Y)| \leq \max\{ |\langle X^{-1}, X-Y \rangle |, |\langle Y^{-1}, X-Y \rangle |  \}$, we can assume either $|\log det (X) - \log det(Y)| \leq  |\langle X^{-1}, X-Y \rangle |$ or $|\log det (X) - \log det(Y)| \leq  |\langle Y^{-1}, X-Y \rangle |$, either of which lead to the same bound in our proof. We corrected this in our revised paper.
>
> ### **Q3: “report the performance of S-mode and B-mode separately”**
> We reported the performance of S-mode and B-mode on both synthetic and real-world datasets and added a related discussion in Section F in the appendix of our revised paper. We observed that the online-like nature of our algorithm tends to reduce the need for B-mode in real-world datasets where the reward functions are often more complex (e.g., less smooth) than the reward functions of synthetic datasets which are usually simpler and smoother.
>
> ### **Q4: “are there any other reasons to prevent full gradient descent training?”**
>
> Another major reason is that it reduces the computational complexity from $\mathcal{O}(n^2)$ in NeuralUCB to $\mathcal{O}(n)$ in NeuraLCB as the former re-trains a new network completely from scratch at each iteration $t$ while the SGD based incremental training adaptively improves one single network sequentially upon the received data.
>
> ### **Q5: Experiments on dependent data**
> Thank you for the suggestion! We have now added the experimental results for dependent data in Section F.1 in the appendix of our revised paper. In summary, we observed that the baseline methods with linear models still highly underperform for the dependent data case while NeuralGreedy shows a significant performance degradation. On the other hand, our method still significantly outperforms the baseline methods in these experiments.

---

> > ### Comment · Reviewer_5NdR · 2021-11-29
> > **Thanks for the response and experiments**
> >
> > Thanks for the response and experiments. Most of my concerns are answered and resolved. It is interesting to observe the different performances between S-mode and B-mode, although I did not fully follow the authors' explanation on the performance difference.
> > While there is a shared concern on the technical novelty and contribution in the reviews, I think this paper should be interesting to the readers and this is a timely result considering the fast advances in pessimism in offline RL.
> > I will keep my original score (6).

---

> > > ### Author Response · Authors · 2021-12-01
> > > **Thank you very much for acknowledging our work**
> > >
> > > We thank you for acknowledging our response and revision, and for appreciating our work.

---

### Official Review · Reviewer_Fwh8 · 2021-11-04

**Correctness:** 4
**Technical Novelty And Significance:** 2
**Empirical Novelty And Significance:** 2
**Recommendation:** 6
**Confidence:** 3

**Main Review:**

Some references are repeated in the bibgraph. For example, there are multiple entries for the same papers (Is pessimism provably efficient for offline rl, Gradient descent provably optimizes over-parameterized neural networks, Neural contextual bandits with ucb-based exploration). In the related work, there is another recent paper that studies policy learning in contextual bandit using the same neural network structure as in this paper (Xu, P., Wen, Z., Zhao, H., & Gu, Q. (2020). Neural contextual bandits with deep representation and shallow exploration. arXiv preprint arXiv:2012.01780.). Similar to the claim of the current work, this paper also improves the computational complexity of neural contextual bandits to a large extent.

In Line 4 of Algorithm 1, are you retrieving the data tuple randomly, sample without replacement, or just in a fixed order? The notation of x_t and x^{(i)}

In the introduction, the authors stated that “actions in the offline data are independent and depend only on the current state” in Rashidinejad et al. (2021). However, it should be also clearly stated that the current paper did not resolve this problem since by Assumption 4.2 we know that the authors in this paper still need the actions to be independent from each other.

Under Assumption 4.2, the authors claimed that the data coverage condition is only imposed on the observed feature contexts. However, a significant difference in the setting of this paper from others that rely on uniform coverage is that the total number of arm contexts are fixed as nK (shown in Assumption 4.1), while other papers such as Nguyen-Tang et al., (2021) do not require this condition. Is this the crucial reason for the relaxation of the condition?

In Theorem 1, it is claimed that the network width m is a polynomial in the number of data n. But it is unclear to me what the exact dependence of m on the number of data n is. In addition, did you also validate the dependency in the experiment settings (m=100, T=15000)?


=========after the authors' response========
I am satisfied with the responses to my questions. I agree that this paper makes good progress in connecting neural contextual bandits and offline settings. I am willing to increase my score for this paper.

**Summary Of The Paper:**

This paper proposes a neural network based contextual bandit algorithm in the offline setting where a dataset of contexts and rewards are given by a logging policy. The goal of the proposed algorithm is to learn an optimal policy from the offline dataset. The proposed algorithm NeuralLCB is similarly structured as the NeuralUCB algorithm (Zhou et a., 2019) in the online setting. The difference is that it uses a lower confidence bound for estimating the reward function instead of an upper confidence bound, and that the optimization procedure for learning the neural network representation is based on the loss on one data point instead of the whole historical data. The authors established an upper bound of the optimal gap of the learned policy and evaluated the algorithm on both simulation and UCI datasets.


**Summary Of The Review:**

My recommendation is mainly based on the theoretical novelty and the applicability of the theory.

---

> ### Author Response · Authors · 2021-11-16
> **Response**
>
> Thank you for your comments. Below we address all your concerns. Please kindly let us know if you have any further questions.
>
> ### **Q1: Missing [Xu, P., Wen, Z., Zhao, H., & Gu, Q. (2020)]**
>
> Thank you for bringing this interesting work to our attention. [Xu, P., Wen, Z., Zhao, H., & Gu, Q. (2020)] proposes to explore only on the last linear layer with an adaptive feature extractor which also significantly reduces the computational complexity of [Zhou et al., 2019]. Our contribution in terms of reducing the computational complexity is orthogonal to [Xu, P., Wen, Z., Zhao, H., & Gu, Q. (2020)] where we focused more on improving the optimization aspect of neural bandits (applicable to both online and offline settings) than on a new neural representation. We added this reference to our revised paper.
>
> ### **Q2: “are you retrieving the data tuple randomly, sample without replacement, or just in a fixed order?”**
>
> Data is retrieved in the exact order as the offline data, sequentially from $(x_1, a_1, r_1)$ to $(x_n, a_n, r_n)$, as we allow the offline action $a_t$ to be dependent on $\mathcal{D}_{t-1}$ during the data collection process.
>
> ### **Q3: “However, it should be also clearly stated that the current paper did not resolve this problem since by Assumption 4.2 we know that the authors in this paper still need the actions to be independent from each other”**
>
> We would like to emphasize Assumption 4.2 does not require actions $a_t$ to be independent of each other. That said, $a_t$ is allowed to be dependent on the past data $\mathcal{D}_{t-1}$. The only independence assumption we made in Assumption 4.2 is that the full contexts $x_t$ are independent of any policy, which standard and minimal in contextual bandits (e.g., $x_t  \overset{i.i.d}{\sim} \rho$).
>
> ### **Q4: “Is this [finite number of arms] the crucial reason for the relaxation of the condition [the empirical single-policy concentration assumption]?”**
>
> The finite number of arms is not crucial for the relaxation of the existing data coverage assumption, though it simplified the setting for more convenience. What’s crucial for such relaxation is the online-like learning of our algorithm, that is, to the best of our knowledge, novel and different than any existing offline learning algorithm in the literature, including Nguyen-Tang et al., (2021). Specifically, conditioned on $\mathcal{D}_{t-1}$, $x_t$, due to pessimism, a dominant term of the regret decomposition is bounded by the expectation of a confidence width w.r.t. the optimal policy $\pi (.| x_t)$
>
> which in turn is bounded by $\kappa$ times the expectation of the confidence width w.r.t. the behaviour policy $\mu(.| \mathcal{D}_{t-1}, {x}_t)$ (please see the proof of Lemma A.2 in the appendix).  This is a key technical reason for the relaxation of the data coverage assumption, which is possible due to the online-like nature of our algorithm. The analysis in [Nguyen-Tang et al., (2021)] or other offline learning algorithms cannot leverage this relaxation because they use the entire batch data to compute an estimated policy.
>
> ### **Q5: “the exact dependence of $m$ on the number of data $n$”**
> The exact dependence of $m$ on $n$ used in our proof is $m = \tilde{\mathcal{O}}(n^{10})$ (see the proof of Lemma 4.1). As we also emphasized in the experiment section that this overparameterization is difficult to realize in practice, we used a neural network with a fixed size in all our experiments ($m=20$ in synthetic datasets and $m=100$ in the real-world datasets). We remark that the goal of our experiments was not to verify such overparameterization, which we believe to be a difficult problem of independent interest even in the original NTK setting. Here, our goal was to verify the empirical effectiveness of our algorithmic design choices in terms of pessimism, optimization, and generalization. A similar fixed size network was also used in the Neural-UCB paper and other papers that are based overparametrized neural networks.

---

> > ### Author Response · Authors · 2021-12-01
> > **Thank you - Please acknowledging our response and revision**
> >
> > Dear Reviewer Fwh8,
> >
> > We would like to particularly thank you again for your valuable comments. As we have put the very effort in addressing all your comments appropriately and promptly, we would very much appreciate it if you could acknowledge our response and revision and update your assessments or score accordingly.
> >
> > Best Regards,
> > Authors

---

> > > ### Comment · Reviewer_Fwh8 · 2021-12-01
> > > **Thank you for your response**
> > >
> > > Thank you for your response. I have read them and other reviews as well. I intended to increase my score for this paper. I am satisfied with your response.

---

> > > > ### Author Response · Authors · 2021-12-01
> > > > **Thank you for your acknowledgement**
> > > >
> > > > Thank you for appreciating our work and increasing your score.

---

### Author Response · Authors · 2021-11-16
**General Response**

Dear our Reviewers and ACs,

Below we addressed all the Reviewers’ concerns.

We have also revised our paper accordingly based on our Reviewers’ suggestions and discussion. The added texts are clearly highlighted in red in our revised version. Specifically, our revisions include:
* Removing repeated references and adding a missing reference, suggested by Reviewer Fwh8
* Adding the experimental results (and discussions) on the dependent data, and reporting the performance of S-mode and B-mode separately in Appendix F of our revised paper, suggested by Reviewer 5NdR
* Adding the experimental result on MNIST in Section 6.2, to further support the suggestion of Reviewer 5NdR
* Correcting a minor mistake in our proof of Lemma B.5, suggested by Reviewer 5NdR
* Adding a clarification on the optimization of NeuraLCB at Section 4 where our optimization design and guarantee are of independent interest that does not only apply to our offline setting but also the online setting of NeuralUCB, from the discussion with Reviewer 5NdR

We would like to clarify that all the revisions are still within the scope of our original paper without changing any conclusions.

Please kindly let us know if you have any further questions. We are more than happy to answer any questions relating to this work.

Thank you very much.

Best Regards,
Authors

---

> ### Author Response · Authors · 2021-11-28
> **Please update any changes in your assessments or score**
>
> Dear our Reviewers,
>
> Thank you again for your valuable comments! We have made the very effort to address all the comments in our reviews appropriately and promptly. As there is only one day left, we would very much appreciate it if after reading our response and revision, you could update any changes in your assessments or score.
>
> Best Regards,
> Authors

---

### Decision · Program_Chairs · 2022-01-20

**Decision:**

Accept (Poster)

**Comment:**

This paper studies off-policy learning of contextual bandits with neural network generalization. The proposed algorithm NeuraLCB acts based on pessimistic estimates of the rewards obtained through lower confidence bounds. NeuraLCB is both analyzed and empirically evaluated.

This paper received four borderline reviews, which improved during the rebuttal phase. The main strengths of this paper are that it is well executed and that the result is timely, considering the recent advances in pessimism for offline RL. The weakness is that the result is not very technically novel, essentially a direct combination of pessimism with neural networks. This paper was discussed and all reviewers agreed that the strengths of this paper outweigh its weaknesses. I agree and recommended this paper to be accepted.